# Bi-fated tendon-to-bone attachment cells are regulated by shared enhancers and KLF transcription factors

**Shiri Kult[1], Tsviya Olender[1], Marco Osterwalder[2,3], Svetalana Markman[1], Dena Leshkowitz[4], Sharon Krief[1], Ronnie Blecher-Gonen[5], Shani Ben-Moshe[6], Lydia Farack[6], Hadas Keren-Shaul[4], Tomer-Meir Salame[4], Terence D Capellini[7], Shalev Itzkovitz[6], Ido Amit[5], Axel Visel[2,8,9], Elazar Zelzer[1]***

[1]Department of Molecular Genetics, Weizmann Institute of Science, Rehovot, Israel; [2]Environmental Genomics and Systems Biology Division, Lawrence Berkeley National, Berkeley, United States; [3]Department for BioMedical Research (DBMR), University of Bern, Bern, Switzerland; [4]Life Sciences Core Facilities, Weizmann Institute of Science, Rehovot, Israel; [5]Department of Immunology, Weizmann Institute of Science, Rehovot, Israel; [6]Department of Molecular Cell Biology, Weizmann Institute of Science, Rehovot, Israel; [7]Department of Human Evolutionary Biology, Harvard University, Department of Human Evolutionary Biology, United States; Broad Institute of Harvard and MIT, Cambridge, United States; [8]U.S. Department of Energy Joint Genome Institute, Lawrence Berkeley National Laboratory, Berkeley, United States; [9]School of Natural Sciences, University of California, Merced, Merced, United States

*For correspondence:
eli.zelzer@weizmann.ac.il

**Competing interests:** The authors declare that no competing interests exist.

**Abstract** The mechanical challenge of attaching elastic tendons to stiff bones is solved by the formation of a unique transitional tissue. Here, we show that murine tendon-to-bone attachment cells are bi-fated, activating a mixture of chondrocyte and tenocyte transcriptomes, under regulation of shared regulatory elements and Krüppel-like factors (KLFs) transcription factors. High-throughput bulk and single-cell RNA sequencing of humeral attachment cells revealed expression of hundreds of chondrogenic and tenogenic genes, which was validated by in situ hybridization and single-molecule ISH. ATAC sequencing showed that attachment cells share accessible intergenic chromatin areas with either tenocytes or chondrocytes. Epigenomic analysis revealed enhancer signatures for most of these regions. Transgenic mouse enhancer reporter assays verified the shared activity of some of these enhancers. Finally, integrative chromatin and motif analyses and transcriptomic data implicated KLFs as regulators of attachment cells. Indeed, blocking expression of both *Klf2* and *Klf4* in developing limb mesenchyme impaired their differentiation.

## Introduction

The function of the musculoskeletal system relies on the proper assemblage of its components, namely skeletal tissues (bone, cartilage, and joints), muscles, and tendons. However, the attachment of tissues composed of materials with large differences in their mechanical properties is highly challenging. In the musculoskeleton, elastic tendons, which have a Young's modulus (a measure of stiffness) in the order of 200 megapascal, are attached to the much harder bone, with a modulus in the order of 20 gigapascal. This disparity makes the connection between these two tissues a mechanical weak point, which is subject to higher incidence of tearing by both external and internal forces acting on the musculoskeleton during movement. The evolutionary solution to this problem is the

enthesis, a transitional tissue that displays a gradual shift in cellular and extracellular properties from the tendon side through to the bone side (*Genin et al., 2009*; *Liu et al., 2011*; *Liu et al., 2012a*; *Lu and Thomopoulos, 2013*; *Thomopoulos et al., 2003*). Yet, despite its importance, the formation of this cellular gradient as well as the underlying molecular mechanism remain largely unknown.

In recent years, the initial events that lead to the formation of the embryonic attachment unit (AU), which serves as the primordium of the enthesis, have started to be investigated. These studies identified the progenitors of the AU and showed that they express both the chondrogenic and teno-genic transcription factors *Sox9* and scleraxis (*Scx*), respectively (*Blitz et al., 2013*; *Sugimoto et al., 2013*). The patterning of the *Sox9*⁺/*Scx*⁺ progenitors along the skeleton is regulated by a genetic program that includes several transcription factors (*Eyal et al., 2019*). Next, *Sox9*⁺/*Scx*⁺ progenitors differentiate to chondrocytes, which form a bone eminence on one side, or to tenocytes, which form the tendon on the other side, whereas the cells in between differentiate into *Gli1*⁺ cells that eventually will form the enthesis (*Felsenthal et al., 2018*; *Schwartz et al., 2017*; *Schwartz et al., 2015*).

Both molecular and mechanical signals regulate the AU. TGFβ signaling regulates the specification of AU progenitors, whereas BMP and FGF signaling as well as mechanical signals determine their fate and differentiation (*Blitz et al., 2013*; *Blitz et al., 2009*; *Roberts et al., 2019*). Postnatal enthesis cells have been termed fibrocartilage cells based on their histological appearance, because they display morphological features that are shared with tenocytes and chondrocytes (*Thomopoulos et al., 2010*). In recent years, several studies have identified some of the genes that these cells express, including collagens type I, II, and X; Indian hedgehog (*Ihh*); parathyroid hormone-related peptide (*Pthlh*); patched 1 (*Ptc1*); runt-related transcription factor 2 (*Runx2*); tenascin C (*Tnc*); and biglycan (*Bgn*) (*Liu et al., 2012b*; *Liu et al., 2013*; *Liu et al., 2018*; *Thomopoulos et al., 2010*). Interestingly, these genes are also expressed by cells in the neighboring tissues, namely by chondrocytes or tenocytes. However, despite these advances, a comprehensive molecular signature of this tissue and the mechanism that enables its formation are still missing.

In this work, we aimed to decipher the identity of the fibrocartilage cells that form the attachment tissue between tendon and bone. Bulk and single-cell transcriptomic analyses of the attachment cells, which were validated by in situ hybridization (ISH) and single-molecule fluorescent ISH (smFISH), showed that these cells express a mix of the transcriptomes of chondrocytes and teno-cytes. Chromatin analysis further verified the transcriptomic results and provided a mechanistic explanation for the bi-fated behavior of attachment cells, which share enhancers with their neighboring tenocytes or chondrocytes. Finally, we identify the transcription factors KLF2 and KLF4 as regulators of attachment cell differentiation. Overall, we provide the transcriptional as well as the epigenetic mechanism that allows attachment cells to activate a combination of cartilage and tendon transcriptomes and, thereby, the formation of the unique transitional tissue.

## Results

### Attachment cell transcriptome is a mix of chondrocyte and tenocyte transcriptomes

To date, the transcriptome of attachment cells has not been characterized thoroughly. We therefore analyzed the transcriptome of embryonic day (E) 14.5 attachment cells from the prominent deltoid tuberosity and greater tuberosity of the humerus. With the goal to isolate these cells specifically, we generated a compound mouse by crossing three mouse lines, namely *Col2a1-Cre*, *R26R-tdTomato*, and *Scx-GFP* (see Materials and methods) (*Blitz et al., 2013*; *Sugimoto et al., 2013*). Thus, the fluorescent reporter tdTomato labeled *Col2a1*-expressing chondrocytes, whereas GFP fluorescently labeled *Scx*-expressing tenocytes. Unexpectedly, the two reporters failed to label the attachment cells that were located in between these two populations. This failure might be due to a missing regulatory element in one of the constructs that were used to produce each transgenic reporter. Nevertheless, the borders between tendon and attachment cells and between cartilage and attachment cells were clearly demarcated (*Figure 1A*, *Figure 1—figure supplement 1A,B*). We therefore used laser capture microdissection (LCM) to subdivide the attachment site into three cellular compartments, namely attachment cells, adjacent tenocytes, and adjacent chondrocytes. As controls, samples were also taken from two more compartments, remote tenocytes and remote chondrocytes. Initial analysis of the different transcriptomes using principal components analysis (PCA) showed that

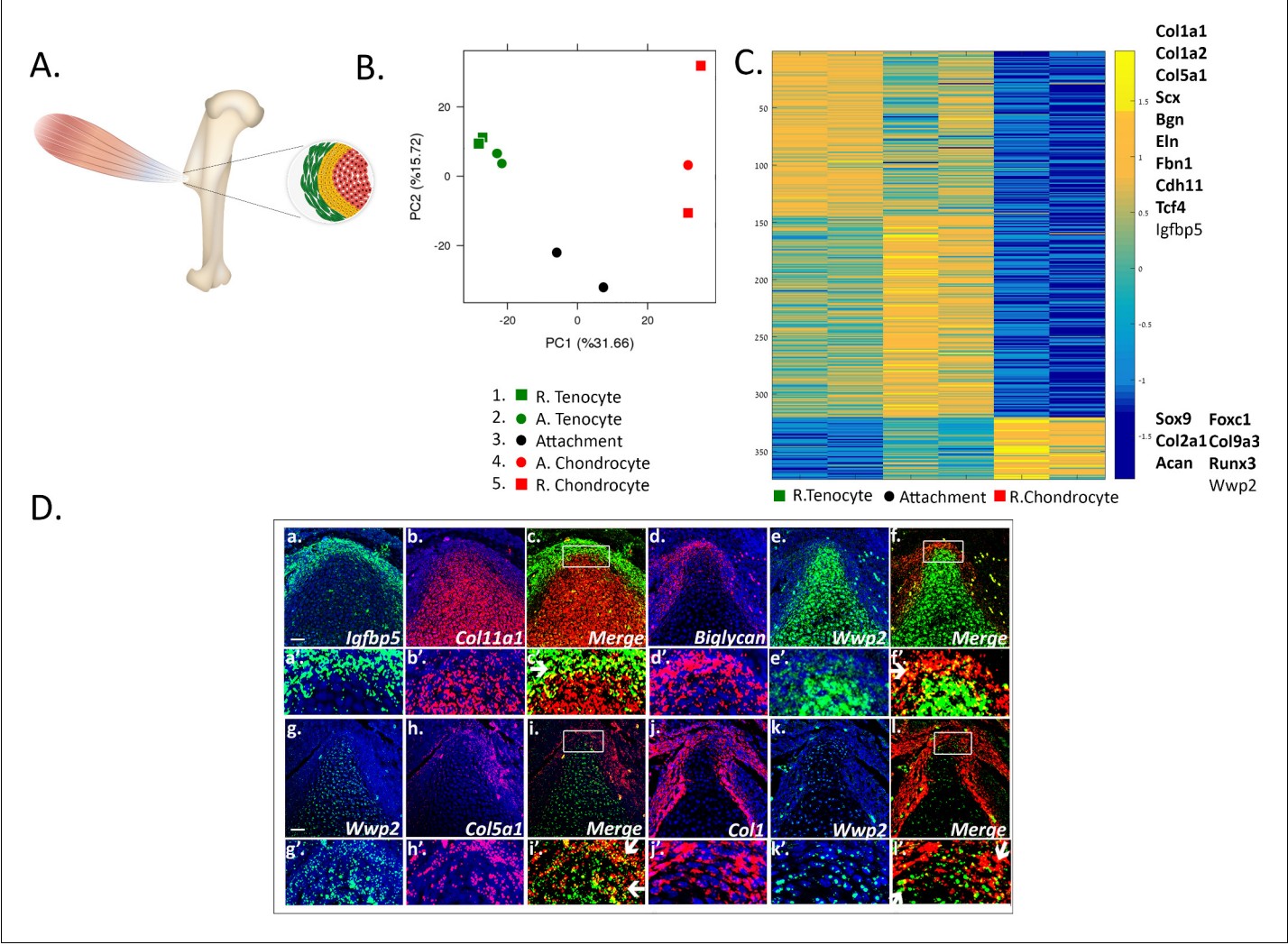

**Figure 1.** Transcriptomic analysis of tendon-to-bone attachment site domains at E14.5. (A) A scheme of tendon-to-bone attachment site of the murine deltoid tuberosity. (B) Principal component analysis (PCA) of bulk MARS-seq data from E14.5 attachment site samples. The *x*-axis (PC1) shows the highest variance among the samples. Interestingly, the samples are arranged according to their anatomical locations. 'R' (samples 1 and 5) stands for remote and 'A' (samples 2 and 4) is for adjacent. The *y*-axis (PC2) shows that tenocytes and chondrocytes are closer to one another, while attachment cells (black circle) were found to be remote from both of them, that is with higher variance, suggesting a unique gene expression profile. (C) Heatmap of gene expression profiles at E14.5 shows 374 selected genes that exhibited differential expression between tenocytes and chondrocytes and were also expressed by attachment cells. Color bar (−1.5-0-1.5) represents the log-normalized counts standardized per gene, as yellow is higher than the mean (0) and blue is lower than the mean. Attachment cells display a gradient of gene expression profiles, reflecting their function as a transitional tissue. The upper cluster contains genes highly expressed in tenocytes (e.g. *Co1a1, Col1a2, Col5a1, Scx, Bgn*), whereas the lower cluster contains genes highly expressed in chondrocytes (e.g. *Sox9, Col2a1, Acan*). Top list on the right contains genes found to be expressed in attachment cells and in tenocytes, whereas bottom list contains genes expressed in attachment cells and chondrocytes; genes in bold type are known tenocyte or chondrocyte markers. D (a-l). Double-fluorescent ISH for mRNA of tendon (*Igfbp5*, biglycan, *Col5a1, Col1a1*) and cartilage (*Col11a1* or *Wwp2*) genes shows that attachment cells (in yellow, shown by arrows) exhibit an in vivo gene expression profile that combines tendon and cartilage genetic programs. a-l: X20 magnification, scale bar: 50 μm; a'-l': magnification of upper panels.

The online version of this article includes the following figure supplement(s) for figure 1:

**Figure supplement 1.** Analysis of the embryonic tendon-to-bone attachment site in a triple-transgenic mouse line.

**Figure supplement 2.** Transcriptomic analysis of tendon-to-bone attachment site domains at E14.5.

**Figure supplement 3.** Upregulated gene expression in attachment cells.

**Figure supplement 4.** Validation of RNA sequencing results by fluorescent in situ hybridization analysis.

the transcriptomes of tenocytes and chondrocytes were clearly separated, whereas attachment cells were located between the two cell types, recapitulating their anatomical positions. This suggests that the attachment cell transcriptome is shared with both chondrocytes and tenocytes (*Figure 1B*, PC1 31.66%).

To further support our initial observation that the transcriptome of the attachment cells is a mixture of chondrocyte and tenocyte transcriptomes, we clustered the statistically significant differentially expressed genes between all samples into five clusters, using CLICK (*Figure 1—figure supplement 2* and see Materials and methods). Out of 865 identified genes, 735 genes were found in two clusters. The first cluster contained mainly known tenogenic genes and the second contained chondrogenic genes. From these two clusters, 374 genes, 320 of them tenogenic and 54 chondrogenic, were also found to be expressed by attachment cells. They included major regulators and marker genes of the two tissues, such as *Sox9, Col2a1*, and *Acan* for chondrocytes and *Col1a1, Col1a2, Scx*, and *Col5a1* for tenocytes (*Figure 1C*).

The third cluster comprised 54 genes that were found to be upregulated in cartilage adjacent to attachment cells alone (*Figure 1—figure supplement 2*). Interestingly, our analysis identified 24 and 23 genes that were found to be down- or upregulated in attachment cells, shown by the fourth and fifth clusters, respectively (*Figure 1—figure supplement 2*). The genes that were found to be uniquely upregulated in attachment cells included transcription factors, such as the Krüppel-like factors (KLFs), *Lmo1*, and *Gli1*, which could act as regulators of the genetic program of attachment cells. In addition, this set included differentiation markers such as *Thy1*, regulators of bone for example *Acp5* and *Alpl*, protein kinases such as *Mapk12* and *Mast2*, and signaling molecules such as *Nod, Traip, Aplnr* and others (*Figure 1—figure supplement 3A,B*). GO analysis of these genes yielded terms relating to regulation of cytokine and IL-12 production, as well as response to laminar fluid shear stress (*Figure 1—figure supplement 3C*). These results clearly show that the transcriptome of the attachment cells includes a mixture of tenocyte and chondrocyte genes, many of which are involved in ECM organization and developmental processes, in addition to a unique subgroup of genes that are upregulated in these cells. To determine the relative proximity of attachment cells to the other two cell types, we performed hierarchical clustering on the 865 differentially expressed genes. Results showed that these cells are closer to tenocytes (*Supplementary file 1*).

To validate our transcriptome analysis, we performed single- and double-fluorescent in situ hybridization (FISH) using marker genes for tenocytes and chondrocytes that were selected from the transcriptomic results, namely *Igfbp5*, biglycan (*Bgn*), *Col5a1*, and *Col1a1* for tenocytes and *Col11a1* and *Wwp2* for chondrocytes. As seen in *Figure 1D* and *Figure 1—figure supplement 4*, in agreement with the transcriptome analysis, the selected markers were co-expressed by the attachment cells.

## Single-cell RNA-seq reveals the bi-fated nature of attachment cells

Finding that the bulk transcriptome of attachment cells represents a mixture of tenocyte and chondrocyte genes led us to examine if this phenotype also exists at a single-cell resolution by performing scRNA-seq. To isolate E13.5 *Sox9*$^+$/*Scx*$^+$ attachment progenitors, we generated a compound mouse line harboring *Sox9-CreER, tdTomato* and *Scx-GFP* transgenes (*Figure 1—figure supplement 1C–E*). Then, FACS-sorted attachment progenitors from the proximal side of the forelimb underwent single-cell 10x Genomics RNA sequencing (*Figure 2A,B* and see Materials and methods). Sequenced progenitors were then filtered, normalized, and clustered (Seurat 3.1.5).

Results showed five transcriptionally distinct subpopulations (*Figure 2A*). Examination of the top-10 differentially expressed genes across clusters revealed that they were subdivided to three main categories. Clusters 1 and 3 were highly rich with chondrogenic genes (*Sox9, Sox5, Col2a1, Wwp2*), whereas clusters 2 and 4 were highly rich with tenogenic genes (*Scx, Hic1, Ptn, Igfbp5*). By contrast, the identity of cluster 0 was less specified, as it was highly rich with genes such as *Col1a1, Alpl, Spp1*, and *Runx2* in addition to *Bicc1, Maf*, and *Mef2c* (*Figure 2—figure supplement 1*).

To study the bi-fated nature of attachment cells, we analyzed the relative expression of tenogenic (*Scx, Col1a2*, biglycan) and chondrogenic (*Sox9, Col2a1, Wwp2*) marker genes in each subpopulation. As seen in *Figure 2B*, all five clusters contained both chondrogenic and tenogenic marker genes. However, while tenogenic genes were relatively high in all five clusters, the expression levels of chondrogenic markers were more variable, being the lowest in cluster 2. Overall, the scRNA-seq

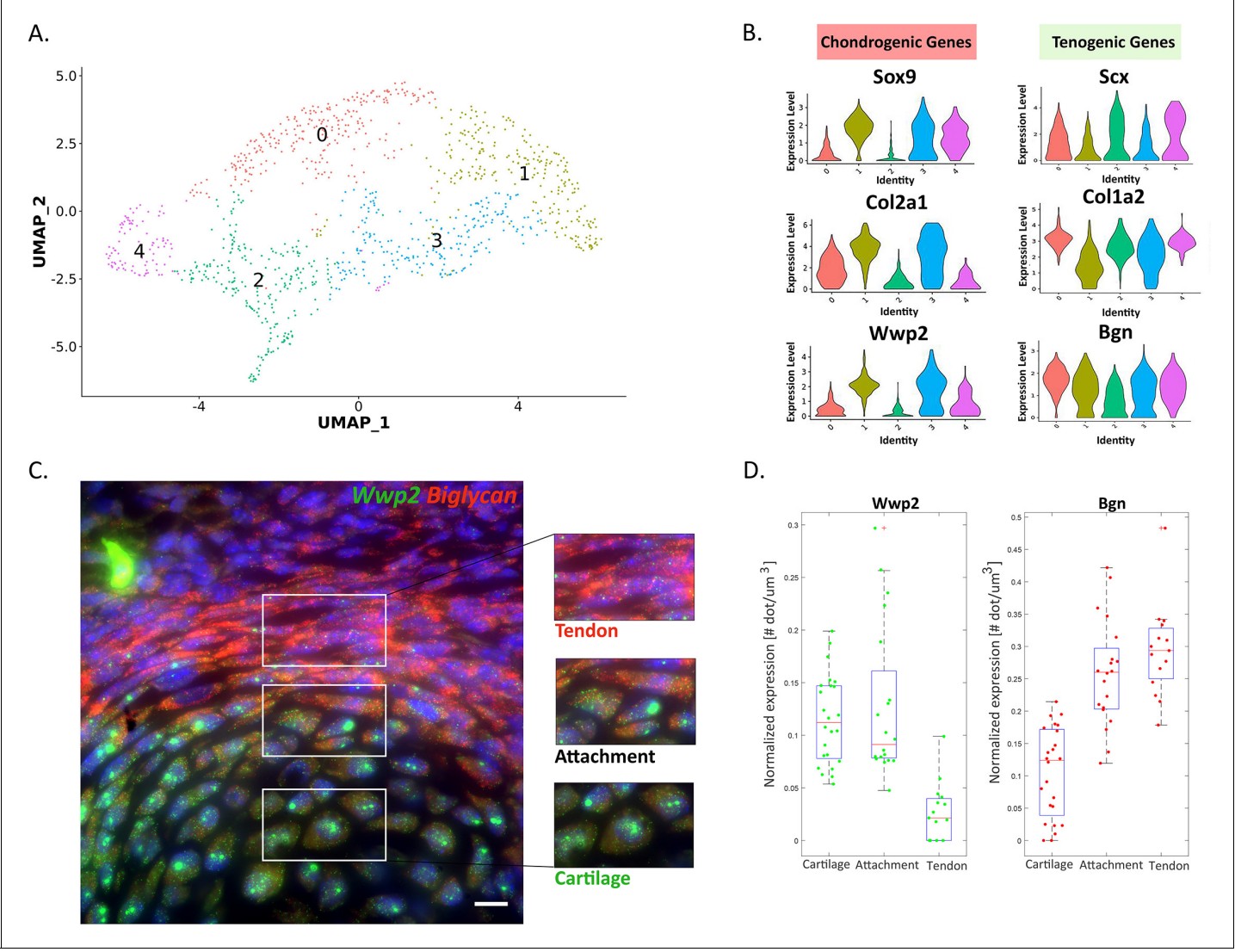

**Figure 2.** Attachment cells co-express tendon and cartilage genes at the single-cell level. (**A**) scRNA-seq analysis of E13.5 Sox9+/Scx+ attachment progenitors is shown as uniform manifold approximation and projection (UMAP) embedding of jointly analyzed single-cell transcriptomes (10x Chromium platform). (**B**) Violin plots of distinct genes associated with tendon or cartilage. (**C**) Single-molecule fluorescent ISH (smFISH) of mRNA of tendon biglycan (Bgn, red) and cartilage Wwp2 (green) genes on the background of DAPI staining (blue) further validates the dFISH results. X100 magnification, scale bar: 10 µm. (**D**) Quantification of Bgn and Wwp2 smFISH results in cartilage, attachment, and tendon cells.

The online version of this article includes the following figure supplement(s) for figure 2:

**Figure supplement 1.** Single-cell RNA-Seq (10x Chromium platform) on E13.5 Sox9+Scx+ attachment progenitors.

results suggest that the *Sox9⁺/Scx⁺* attachment progenitors are bi-fated. The analysis also revealed a molecular sub-classification of this progenitor population.

Finally, to confirm the co-expression of tenogenic and chondrogenic genes at the attachment site at a single-cell resolution, we performed single-molecule FISH for *Wwp2* (chondrogenic marker) and *Bgn* (tenogenic marker). As seen and quantified in ***Figure 2C,D***, *Wwp2* and *Bgn* were indeed co-expressed in the attachment cells.

Overall, these results indicate that attachment cells are bi-fated, expressing in parallel both chondrogenic and tenogenic genes (referred to in the following as mixed transcriptome or mixed expression profile).

## Genome-wide profiling of attachment cell-specific regulatory regions

To gain a mechanistic understanding of how attachment cells activate a combination of two transcriptomes, we compared chromatin accessibility in these cells with open chromatin signatures defining chondrocytes and tenocytes by conducting an assay for transposase-accessible chromatin with high-throughput sequencing (ATAC-seq) (*Buenrostro et al., 2015*). This method allows to profile open chromatin regions, some of which may act as enhancers. To isolate by FACS E13.5 humeral chondrocytes, tenocytes and attachment cells, we generated a compound mouse line harboring *Sox9-CreER*, *tdTomato*, and *Scx-GFP* transgenes (*Figure 1—figure supplement 1C–E*). However, because the number of isolated chondrocytes was insufficient, chondrocytes were FACS-sorted from E13.5 *Col2a1-CreER^T^-tdTomato-Scx-GFP* mouse. These three cell populations were then subjected to ATAC-seq (see Materials and methods).

Initial PCA analysis of accessible chromatin profiles for each FACS-sorted cell population once again revealed that tenocytes and chondrocytes were clearly separated, while attachment cells resided between these two cell types (*Figure 3—figure supplement 1A*, *Figure 1B*). Next, we compared global chromatin accessibility among the three cell types by calculating the level of overlap among the ATAC-seq peaks (*Figure 3A*). While the majority of the peaks were shared by all three cell types, attachment cells had a significantly lower number of unique peaks (p<1e-4 relative to both cell types, chi-square with Yates correction), and a significantly higher overlap with the other two cell types (p<2.2e-1).

Analysis of the ATAC-seq signal revealed that 13,017 peaks were located near transcription start sites (TSSs), whereas 31,856 peaks were in intergenic or intron regions. Most of the peaks that were located near TSSs were accessible in all three cell types (87%), and only 13% were accessible in one or two cell types (*Figure 3—figure supplement 1C,D*).

Next, we studied the ATAC-seq signal of peaks associated with the genes that were differentially expressed at E14.5, using HOMER default parameters. We found that 819 peaks were located near transcription start sites (TSSs), whereas 2340 peaks were in intergenic or intron regions. Most of the peaks that were located near TSSs (708, 86%) were accessible in all three cell types, and only 111 (13%) were accessible in one or two cell types (*Figure 3B,C*). This low level of differential accessibility is inconsistent with the possibility that promoter accessibility is the main mechanism regulating the bi-fated attachment cells. Interestingly, a significantly higher fraction of intergenic peaks were specific to one or two cell types (1767, 74.7%, p=0, chi-square test, *Figure 3B,C*).

Out of these 1767 intergenic peaks, 920 peaks were accessible in attachment cell; 672 were shared with either tenocytes or chondrocytes; and 248 were accessible only in attachment cells (*Supplementary file 3*). These results suggest that the 920 intergenic elements that were accessible in attachment cells may act as enhancers that drive the transcriptome of these cells. Moreover, since most of these intergenic elements were shared between attachment cells and either chondrocytes or tenocytes, they may serve as part of the mechanism that drives the mixed transcriptome of attachment cells.

To identify such dual cell-type-specific enhancers likely regulating attachment cell differentiation, we next screened for shared enhancers of 15 bona fide markers of tenocytes or chondrocytes that were found to be expressed in E14.5 attachment cells (*Figure 1C*). To improve the prediction of these enhancers, we selected our ATAC-seq peaks based on their proximity to genes with verified expression in attachment cells and another cell type (*Figure 1—figure supplement 4*, *Figure 1D*) and computationally intersected them with ENCODE datasets of histone modification marks associated with enhancers and promoters in mouse limbs at E13.5 (H3K27Ac, H3K9ac, H3K4me3, H3K4me1 ChIP-Seq), and other datasets (*Andrey et al., 2017*; *Guo et al., 2017*; *Figure 4* and *Table 1*), revealing the degree of evolutionary conservation of each core sequence (*Casper et al., 2018*). For example, as shown in *Figure 3D*, we identified a region at −42 kb from the TSS of Mgp, a *bona fide* chondrogenic marker (*Barone et al., 1991*), which was accessible in chondrocytes and attachment cells, whereas in tenocytes this site was closed. Another example is *Sox9*, a bona fide chondrogenic marker. At +303 kb from *Sox9*, we identified a region that was accessible in attachment cells and chondrocytes, but not in tenocytes. The same pattern was observed for a region at −330 kb from the TSS of a third bona fide chondrogenic marker, namely *Col11a1* (*Li et al., 1995*). The opposite pattern was observed at −17 kb from the TSS of *Col1a2*, a bona fide tenogenic marker, where we identified a region that was accessible in attachment cells and tenocytes, but not

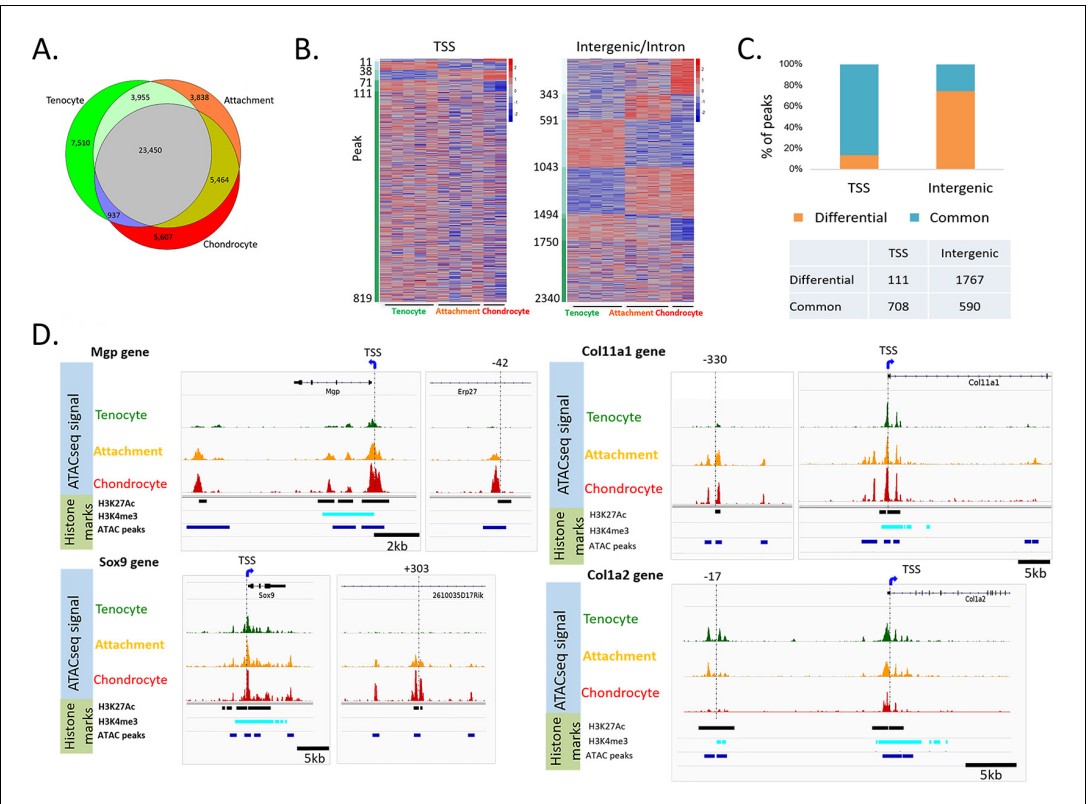

**Figure 3.** Accessible chromatin reveals an epigenetic mechanism shared by attachment cells and neighboring tenocytes or chondrocytes. (**A**) Venn diagram showing cell-specific or overlapping peaks of ATAC-seq among tenocytes, chondrocytes and attachment cells. (**B**) Heatmap of ATAC-seq peaks associated with E14.5 differentially expressed genes. Left: TSS peaks, right: intergenic or intron peaks. The peaks are sorted according to their degree of accessibility across the three cell types. (**C**) Percentage of common peaks (shared by three cell types) vs. differential peaks (the chromatin is open only in one or two cell types) compared between TSS and intergenic areas (p=0, chi-square test). (**D**) IGV snapshots of the TSS region of *Mgp*, *Sox9*, *Col11a1*, and *Col1a2* genes, as well as potential enhancers of these genes.

The online version of this article includes the following figure supplement(s) for figure 3:

**Figure supplement 1.** Accessible chromatin reveals an epigenetic mechanism of shared enhancers by attachment cells and neighboring tenocytes or chondrocytes.

in chondrocytes. Similar results were obtained for additional chondrogenic markers, such as *Sox6*, and for tenogenic markers *Tnc* and *Col1a1* (data not shown). Importantly, we found that the chromatin accessibility patterns of these putative enhancers were in agreement with the transcriptomic and ISH results, as shown, for example, by *Sox9* and *Col5a1* (*Figure 1C,D*, *Figure 1—figure supplement 4*). This suggests that the mechanism for the activation of a mixed transcriptome in attachment cells is based on sharing regulatory elements with chondrocytes or tenocytes.

## Shared regulatory elements drive expression in attachment cells and flanking cartilage or tendon cells

The identification of multiple predicted enhancer regions near genes expressed by attachment cells and tenocytes or chondrocytes suggests that attachment cells are regulated predominantly by enhancers with shared activities. To test this hypothesis in vivo, we took advantage of a recently developed, site-directed transgenic mouse enhancer reporter system (*Kvon et al., 2020*). Using this system, we studied the activity of eight elements, which were selected because they were associated with bona fide marker genes for tenocytes or chondrocytes, and were found to be expressed in the attachment cells. Moreover, they were predicted to drive transcription in attachment cells and one of the flanking tissues (*Figure 3D*). The activity of these representative elements was examined at

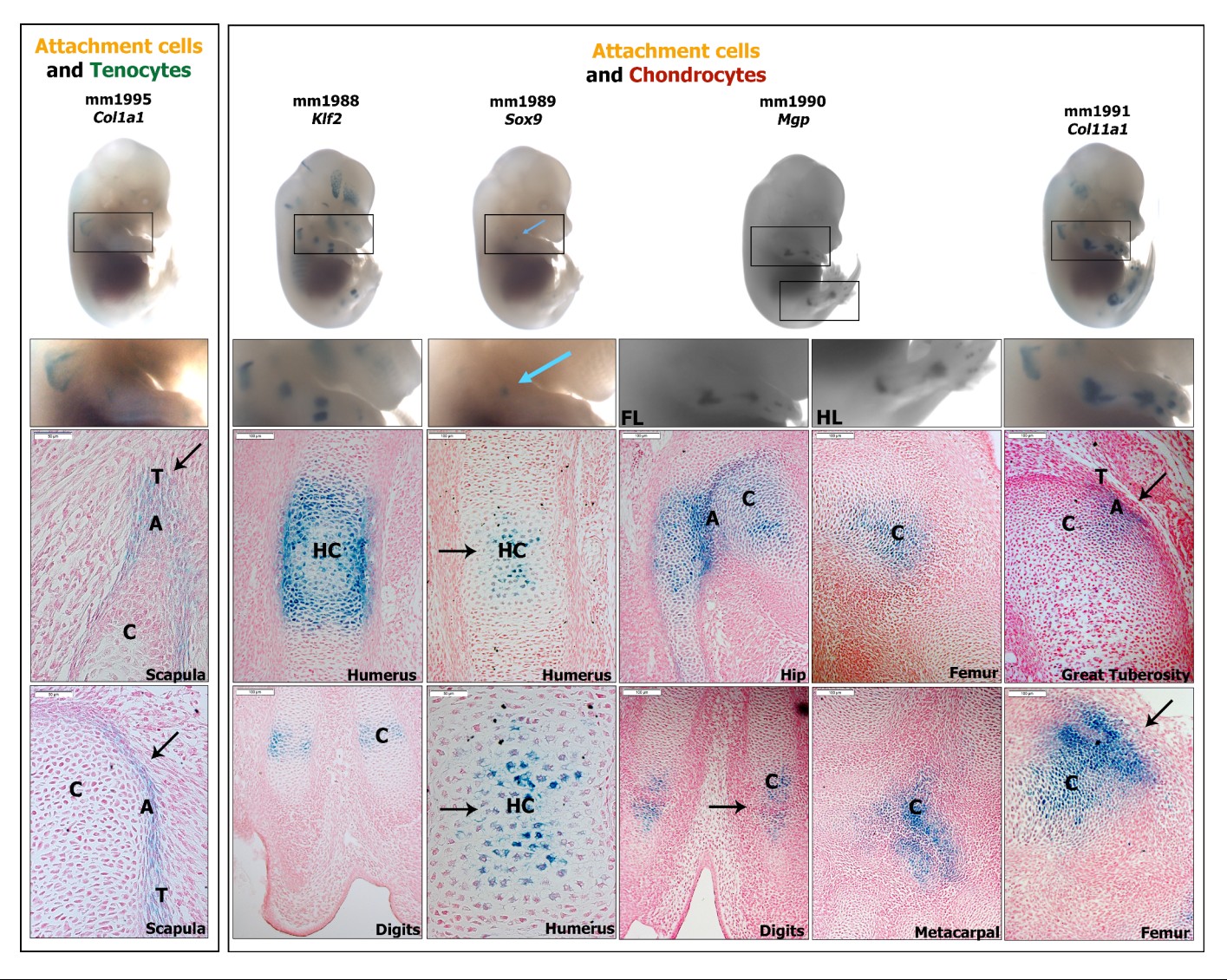

**Figure 4.** In vivo analysis of enhancers identifies shared domains of activity between attachment cells and neighboring tissues. Transgenic mouse reporter enhancer assay (lacZ) of elements positive at E14.5 (marked in light blue; for each enhancer, an E14.5 whole-mount embryo, magnification of the limb and forelimb and/or hindlimb sagittal sections are shown). Left to right: Col1a1 element (mm1995) activity is seen at the teres major insertion at the scapula (n=2/9). Klf2 element (mm1988) activity is seen in hypertrophic chondrocytes and perichondrium at the humerus and forelimb digits (n=3/3). Sox9 element (mm1989) activity is seen in hypertrophic chondrocytes of the humerus (n=4/7). Mgp element (mm1990) activity is seen in the hip, digit, and metacarpals joints in addition to the posterior distal side of the femur (n=3/3). Col11a1 element (mm1991) activity is seen in the greater tuberosity insertion and anterior distal side of the femur (n=11/11). T, tendon; C, cartilage; A, attachment; HC, hypertrophic chondrocytes.

E14.5, a stage at which chondrocytes, tenocytes, and attachment cells have already been established.

Five elements were found to be active in the mouse forelimbs, as well as in other anatomical areas (*Figure 4*). The *Col1a1*-associated element drove *lacZ* expression at the teres major insertion into the scapula, in agreement with the ATAC-seq results, which predicted its activity in tenocytes and attachment cells (n = 2/9). This result suggests that *Col1a1* element is active in tenocytes and attachment cells. *Klf2* and *Sox9* elements were predicted to be active in chondrocytes and attachment cells (*Figure 3D*). The *Klf2* element indeed drove reporter activity in hypertrophic chondrocytes and perichondrium at the humerus and forelimb digits, in addition to the skull and mandible (*Figure 4*, n = 3/3). The *Sox9* element showed activity solely in hypertrophic chondrocytes of the humerus (n = 4/7). These results suggest a chondrocyte-specific function of these two enhancers.

**Table 1.** Criteria that were used to select candidates for the in vivo enhancer activity assay.
We first compared the ATAC-seq data to bulk MARS-seq transcriptome analysis and chose peaks that were assigned to genes with differential expression (using GREAT). According to the ATAC-seq results, the TSS of most of these genes was accessible in all three tissue types. Therefore, we searched for putative enhancers that could regulate the differential gene expression. We then compared the elements overlap with E13.5-E14.5 histone marks (H3K27Ac or H3K4me1 of ENCODE) and HiC results (*Andrey et al., 2017*), to increase the probability of in vivo verification, in addition to preliminary results of gene expression (i.e. the chosen elements were assigned to genes of interest, which were identified by RNA-seq and validated by ISH). The degree of evolutionary conservation of the core sequence was also taken into consideration while prioritizing the elements.

| | mm1995 Col1a1 | mm1988 Klf2 | mm1989 Sox9 | mm1990 Mgp | mm1991 Col11a1 | mm1992, mm1994, Eln | mm1986 Igfbp5 |
|---|---|---|---|---|---|---|---|
| Distance from TSS | −63,010 | −30,343 | 303,066 | −42,288 | −331,264 | −3,433 −43,487 | −3,605 |
| Coordinate | Chr11:94872064– 94874364 | Chr8: 72287698– 72291754 | Chr11:113084290– 113086290 | Chr6:136917343– 136918843 | Chr3: 113698476– 113700076 | Chr5: 134749924– 134751424 Chr5: 134790228– 134791228 | Chr1: 72877526– 72879452 |
| Encode H3K27ac/ H3K4me1 | + | + | + | + | + | + | + |
| HiC | | + | | | | | |
| Core seq conservation | Opossum | Chicken | Platypus | Opossum | Lizard | Opossum | Chicken |
| Klf2 binding site | + | + | + | - | - | | |

*Mgp* element was also predicted to be active in chondrocytes and attachment cells (*Figure 3D*). Its activity was seen in forelimb and hindlimb, specifically in hip, metacarpal joints and digits as well as in the posterior distal side of the femur, a site where ligaments (e.g. the cruciate ligaments) are inserted into the femur at the knee area and at ligament insertion into to the hip (e.g. iliofemoral ligament; *Figure 4*, n = 3/3), verifying its activity in chondrocytes and attachment cells. Lastly, *Col11a1* element activity was predicted in chondrocytes and attachment cells (*Figure 3D*). Its activity verified the bioinformatic analysis, showing LacZ staining in the greater tuberosity insertion, as well as in the posterior side of the skull, the nasal bone area and the anterior distal side of the femur (*Figure 4*, n = 11/11).

These results suggest that *Mgp* and *Col11a1* elements are active in chondrocytes and attachment cells, whereas *Col1a1* element is active in tenocytes and attachment cells, as the chromatin analysis predicts. Overall, these results provide a proof of concept for the ability of the accessible intergenic elements we have identified to act as enhancers that drive expression in both attachment cells and chondrocytes or tenocytes. This supports our hypothesis that shared enhancers activate a mixed transcriptome in attachment cells.

## Krüppel-like factors are regulators of attachment cell development

Our finding of enhancers that can drive the transcription of the mixed transcriptome of the attachment cells raised the question of the identity of the transcription factors (TFs) that can potentially bind to these elements. To identify such factors, we used Genomatix to analyze accessible elements that were associated with differentially expressed genes for over-representation of transcription-factor-binding sites (TFBS), selecting the top 50 TFBS families, and then mining our transcriptomic data for the expression of these TFBS families (see Materials and methods). Among the differentially expressed genes at E14.5 we identified NFIs (*Nfia*), GLIs (*Gli1*), KLFs (*Klf2* and *Klf4*), ZBTBs (*Zbtb48*) and RUNXs (*Runx3*; *Table 2*, *Figure 5A*), whose expression was upregulated in attachment cells. Further support for these results was provided by HOMER motif analysis (*Heinz et al., 2010*), which showed significant over-representation of KLFs and RUNXs TFBSs. We therefore sought to explore the possible role of KLFs as regulators of attachment cells.

Focusing on *Klf2*, which was found to be differentially expressed by the bulk transcriptome analysis of the attachment site (*Figure 1—figure supplement 2*, *Figure 1—figure supplement 3*), we first

**Table 2.** Genomatix analysis of the genomic regions of cis-regulatory elements identified by ATAC-seq.

Over-representation of transcription factor binding site (TFBS) families was identified. Crossing these results with E14.5 transcriptome revealed differentially expressed TFs from the KLF (*Klf4* and *Klf2*), GLI (*Gli11*), NFI (*Nfia*), ZBTBs (*Zbtb48*), and RUNX families.

| TFBS | Gene family | Description | Zinc finger |
|------|-------------|-------------|-------------|
| V$E2FF | Myc | E2F-myc activator/cell cycle regulator | no |
| V$KLFS | KLF | Krüppel-like transcription factors | yes |
| V$ZF02 | Zbtb | C2H2 zinc finger transcription factors | yes |
| V$NF1F | Nfi | Nuclear factor 1 | yes |
| V$MAZF | Maz, Patz1 | Myc-associated zinc fingers | yes |
| V$GLIF | GLI | GLI zinc finger family | yes |
| V$PLAG | Plag1, Plag2, Plagl1 | Pleomorphic adenoma gene | yes |
| V$HAML | Runx | Acute myelogenous leukemia factors | no |
| V$NFKB | HIVEP | Nuclear factor kappa B/c-rel | yes |
| V$SMAD | Smad | Vertebrate SMAD family of transcription factors | no |

validated its expression in the forming attachment site by ISH (*Figure 5B*). Next, we analyzed the enhancers that were shown by the bioinformatic analysis to be active in attachment cells and either tenocytes or chondrocytes (*Figure 4*). Three of these enhancers had Klf2-binding sites in their sequence (*Table 1*), further supporting a potential role for KLFs during attachment site development.

Previous studies demonstrated that Klf2 and Klf4 are functionally redundant, as KLF4 has ~90% sequence similarity to KLF2 in its zinc finger DNA-binding domain, suggesting that these factors could have common target sequences (*Chiplunkar et al., 2013*). In our scRNA-seq analysis, the expression of both genes was found in *Sox9+/Scx+* attachment progenitors (*Figure 5C*). We therefore proceeded to study attachment cell development upon blocking the expression of both *Klf2* and *Klf4* in limb mesenchyme, using *Prx1-Cre* as a deleter and focusing on E15.5 and E18.5, a period during which the attachment site of the deltoid tuberosity undergoes differentiation and consequently grows in size. Transverse histological sections through the deltoid tuberosity of E15.5 control mice showed that the attachment cells were packed together and surrounded by ECM (*Figure 5Da'*). In contrast, in the putative attachment site of *Prx1-Klf2-Klf4* double conditional knock-out (dcKO) embryos, the cells were sparse with reduced ECM (*Figure 5Db'*). By E18.5, this difference was more pronounced, as attachment cells failed to differentiate (*Figure 5Dc',d'*). To gain a molecular understanding, we studied the expression of several genes that were previously shown to be expressed at these stages in the attachment site (*Felsenthal et al., 2018*; *Figure 1D*). Indeed, we found that the expression of *Col1a1*, *Gli1*, *Bsp*, *Bgn*, and *Col5a1* was reduced in the dcKO attachment site, relative to the control (*Figure 5De-p*), supporting a role for KLF2/4 in attachment cells differentiation.

Finally, to further validate the involvement of KLFs in activation of gene expression in the attachment site, we searched for KLF2/4 binding sites in ATAC-seq peaks associated with the 374 genes that were shown to be expressed by attachment cells (*Figure 1C*). Interestingly, we found that many of these genes had KLF2/4 binding sites in their regulatory regions (72% of the 374 attachment genes relative to 53% in the whole genome, p<1e-4, chi-square). We then searched for KLF2/4 binding sites in ATAC-seq peaks that were associated with genes whose expression was reduced in the dcKO attachment site (*Figure 5De-p*). For *Gli1*, we found KLF2/4 TFBSs in peaks that reside −2.1 and −1.5 kb from its TSS (*Table 3*). For *Col5a1*, we found multiple binding sites for KLF2 or KLF4. Together, these results indicate that KLF2/4 play an essential role in regulating attachment cell gene expression.

Combined with the bioinformatic analysis of chromatin and transcriptomic data, these results suggest that KLF2/4 are major regulators of tendon-to-bone attachment, playing a central role in attachment cell differentiation.

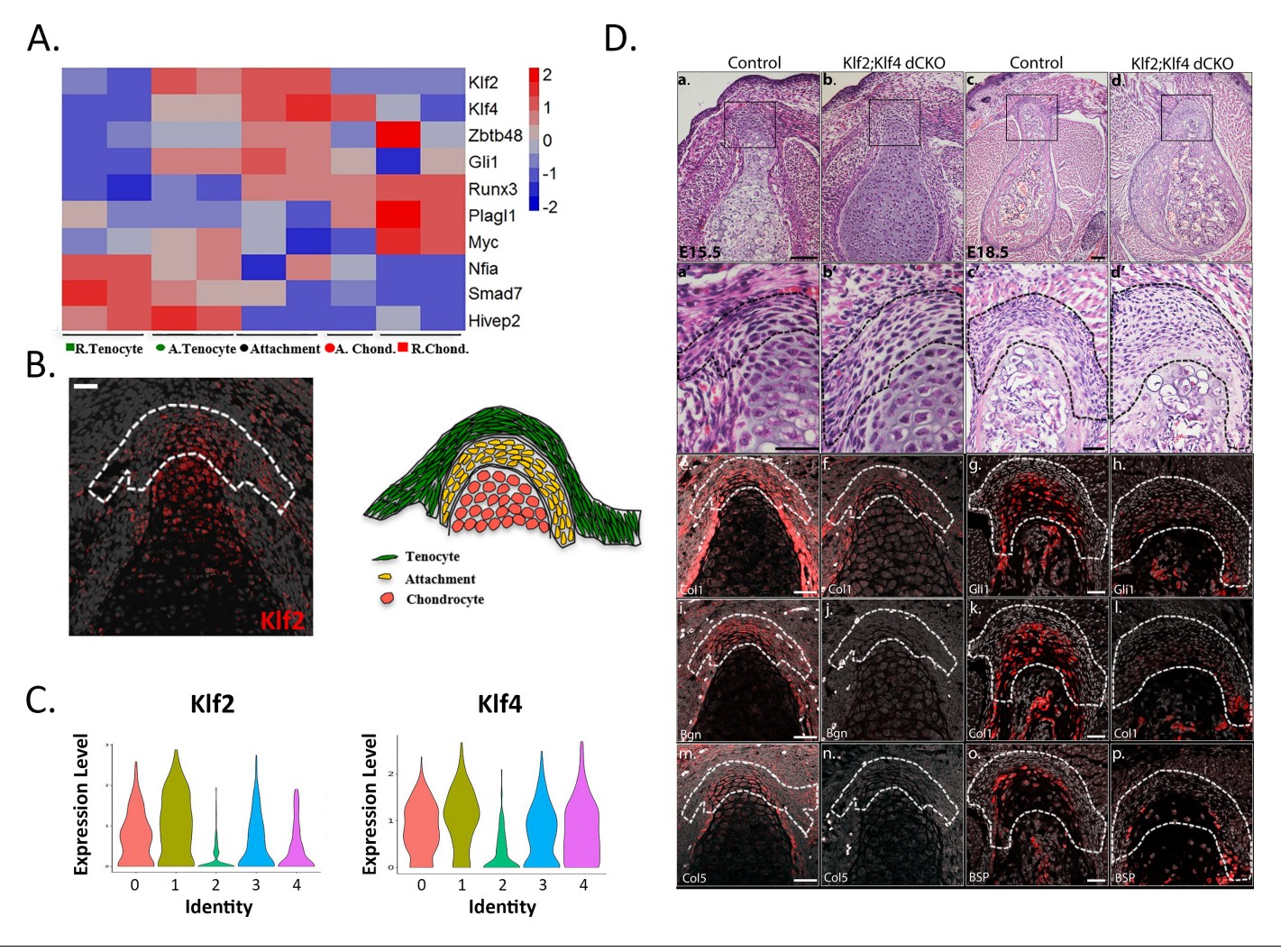

**Figure 5.** Krüppel-like factors (KLFs) are regulators of attachment cell development. (A) Heatmap of selected transcription factors at E14.5. Transcriptome analysis shows upregulated expression of *Klf2, Klf4,* and *Gli1* in attachment cells. (B) Left: E14.5 ISH validated these results, showing *Klf2* expression in attachment cells (X20 magnification, scale bar: 25 μm). Right: Scheme of attachment site. (C) Single-cell RNA-seq analysis of E13.5 *Sox9 +Scx+* attachment cells shows *Klf2* and *Klf4* expression in the five cell populations. (D) KLF2 and KLF4 are regulators of attachment cell development. a-d. Histological transverse sections through the humeral deltoid tuberosity of E15.5 *Prx1-Klf2-Klf4* and E18.5 *Prx1-Klf2-Klf4* mutant and control embryos (×10 and ×5 magnification, scale bar: 100 μm). a′-d′. Higher magnification of upper panel (×40 for E15.5 and ×20 for E18.5, scale bar: 40 μm). e-p. ISH for *Col1a1* and *Bgn* genes of E15.5 *Prx1-Klf2-Klf4* mutant and control embryos. ISH for *Gli1, Col1a1,* and *Bsp* genes of E18.5 *Prx1-Klf2-Klf4* mutant and control embryos (×20 magnification, scale bar: 40 μm).

## Discussion

In this work, we describe the unique transcriptome that allows cells of the attachment between tendon and bone to act as a transitional tissue. The ability to activate a combination of chondrogenic and tenogenic transcriptomes is regulated by sharing enhancers with these cells. Finally, we identify the transcription factors KLF2/4 as regulators of these unique bi-fated cells.

The existence of borders between tissues that differ in cell type, extracellular matrix composition, structure, and function raises the question of how tissues are connected. The border can be sharp, as seen in blood vessels, where pericytes and endothelial cells are separated by a basement membrane, or between the esophagus and the stomach in the gastrointestinal tract (***Bergers and Song, 2005***; ***San Roman and Shivdasani, 2011***). On the other hand, the border can be less defined histologically and molecularly, thus forming a transitional tissue. Examples for the latter are the borders between the sections of the small intestine and between tendon and bone (***Liu et al., 2007***;

**Table 3.** *Gli1* and *Col5a1* genes are expressed in the attachment site and have TFBS for Krüppel-like factors (KLFs).

Search for KLF2/4 binding sites in ATAC-seq peaks that were associated with genes whose expression was reduced in the dcKO attachment site (*Figure 5De-p*). For *Gli1*, we found KLF2/4 TFBSs in peaks that reside −2.1 and −1.5 kb from its TSS (indicated as + under TFBS). For *Col5a1*, we found multiple binding sites for KLF2 or KLF4 (indicated as + under TFBS).

| Peak | HOMER | | GREAT | | | | | | ENCODE | | | TFBS | | |
|---|---|---|---|---|---|---|---|---|---|---|---|---|---|---|
| | Gene | Distance to TSS | Annotation | gene1 | Distance gene 1 | gene2 | Distance gene 2 | H3K27ac | H3K4me1 | H3K4me3 | KLF2 (Genomatix) | KLF4 (Genomatix) | KLF4 (Homer) |
| chr10_127339817_127340317 | Gli1 | 1512 | Intron | Gli1 | 1522 | Arhgap9 | 16340 | + | + | + | + | + | + |
| chr10_127339170_127339670 | Gli1 | 2159 | Intron | Gli1 | 2169 | Arhgap9 | 15693 | + | + | | + | + | + |
| chr10_127338776_127339276 | Gli1 | 2553 | Intron | Gli1 | 2563 | Arhgap9 | 15299 | + | + | + | | | |
| chr10_127341889_127342389 | Gli1 | −560 | TSS | Gli1 | −550 | NA | NA | + | + | + | | | |
| chr10_127342228_127342728 | Gli1 | −899 | TSS | Gli1 | −889 | NA | NA | + | + | + | | | |
| chr2_27740165_27740665 | Rxra | 29832 | Intron | Col5a1 | −146010 | Rxra | 63214 | + | + | + | + | | |
| chr2_27745961_27746569 | Rxra | 35682 | Intron | Col5a1 | −140160 | Rxra | 69064 | + | + | | + | | |
| chr2_27934260_27934760 | Col5a1 | 48085 | Intron | Col5a1 | 48085 | Fcnb | 150375 | + | + | | | + | |
| chr2_27886819_27887319 | Col5a1 | 644 | Intron | Col5a1 | 644 | NA | NA | + | + | + | | | + |
| chr2_27802472_27802972 | Col5a1 | −83703 | Intergenic | Col5a1 | −83703 | Rxra | 125521 | + | | | + | | |
| chr2_27848386_27848886 | Col5a1 | −37789 | Intergenic | Col5a1 | −37789 | Rxra | 171435 | + | + | | + | | |
| chr2_27885904_27886561 | Col5a1 | −192 | TSS | Col5a1 | −192 | NA | NA | + | + | | + | | |
| chr2_27813321_27813821 | Col5a1 | −72854 | Intergenic | Col5a1 | −72854 | Rxra | 136370 | + | + | | | | |

*Romih et al., 2005*; *San Roman and Shivdasani, 2011*). From a broader perspective, as all organs and systems are made of different tissues, understanding the biology of border tissues is imperative. Moreover, some of these border tissues are involved in various pathologies. For example, gastric cancers may emerge from distinct anatomical areas, such as the esophagus–stomach boundary (*Chawengsaksophak, 2019*; *McDonald et al., 2015*; *San Roman and Shivdasani, 2011*).

In the case of the attachment between tendon and bone, the significance of this tissue is demonstrated by enthesopathies, a collective name for injuries and pathologies of the enthesis. For example, over 30% of the population over the age of 60 will injure their shoulder's rotator cuff (*Lehman et al., 1995*). Failure rates of surgical reattachment range from 20% for small tears to 94% for repair of massive tears (*Galatz et al., 2004*; *Harryman et al., 1991*). The high failure and recurrence rates of these procedures highlight the need for understanding the biology of this complex transitional tissue of the enthesis. This understanding may allow the development of new strategies to improve the treatment of enthesopathies.

There are two options to form a transitional tissue. The first strategy is by mixing cells from the two neighboring tissues, such as in the epithelia of the urinary tract (*Romih et al., 2005*). Alternatively, the border cells can express a mixture of the transcriptomes of the two neighboring cell types. As we show here, the attachment cells represent the latter strategy well, as they express a high number of genes that are differentially expressed by either tenocytes or chondrocytes. These cells display morphological features that are shared with tenocytes and chondrocytes (*Thomopoulos et al., 2010*). Our results therefore provide a molecular explanation for the age-old histological definition of enthesis cells as fibrocartilage, which was based on their morphology (*Thomopoulos et al., 2010*). Moreover, the finding of mixed matrix genes in the transcriptome of the attachment cells may provide a mechanism for the formation of a transitional tissue, which allows safe transfer of forces by the tendon between muscle and bone.

In addition to expression of chondrogenic and tenogenic genes, we identified genes that are uniquely expressed by attachment cells. These genes may provide another level of specificity to the regulation of the development of this unique tissue. Finally, our scRNA-seq of the attachment cells revealed heterogeneity, which was a consequence of varying levels of chondrogenic and tenogenic gene expression. This heterogeneity may represent the differentiation processes that the *Sox9/Scx*-positive progenitors undergo during development, ultimately leading to their terminal cell fate.

Our finding that attachment cells are bi-fated raises the question of the mechanism that underlies this fate. An immediate implication of our finding is that there must be an epigenetic mechanism that supports the bi-fated state. The observed chromatin accessibility at the sites of the promoters of most of the shared genes in all three cell types rules out the possibility of limited promoter accessibility as the main mechanism. By contrast, the high percentage of shared accessible intergenic sites between attachment cells and one group of flanking cells, that is chondrocytes or tenocytes, suggests that this is the main mechanism. Moreover, many of these shared sites correlated with putative enhancers in ENCODE datasets. Finally, we verified the ability of three different enhancers from that list to drive gene expression in attachment cells and in either tendon or cartilage. These findings strongly support our hypothesis that the regulatory mechanism is based on the ability of attachment cells to share enhancers with either chondrocytes or tenocytes in order to drive the mixed expression profile of these bi-fated cells. It is important to emphasize that we have also identified a group of intergenic elements that were accessible exclusively in attachment cells. These attachment-specific elements may act as enhancers that drive the expression of genes that are specific to attachment cells. It is, of course, possible that they participate in the regulation of shared genes as well. Overall, both sets of putative enhancers, namely shared and attachment cell-specific, may play important roles in the genetic program that regulates the development of this unique tissue.

Sharing enhancers is not the only possible strategy for the generation of a mixed transcriptome. A simple alternative would be a specific set of enhancers to be used by the attachment cells. A possible explanation for the sharing strategy is the common origin of all these cells, which is limb mesenchyme originating from lateral plate mesoderm (*Johnson and Tabin, 1997*). It is possible that during development, limb mesenchymal progenitors display highly accessible chromatin; yet, during differentiation, this accessibility is restricted to prevent the expression of genes from alternate lineages. In contrast to this restriction process, in the bi-fated attachment cells the shared sites are maintained accessible to allow the expression of the mixed transcriptome. A mechanism for silencing of genes of alternate lineages was previously described. For example, polycomb-repressed chromatin

leads to silencing of genes of alternate lineages, leading to the commitment to a specific cell fate (*Aldiri et al., 2017*; *Laugesen and Helin, 2014*; *Zhu et al., 2013*). Interestingly, previous studies demonstrated the importance of chromatin repression in the developing limb, showing how deletion of *Ezh2*, which acts as the enzymatically active subunit of PRC2, leads to skeletal malformations (*Deimling et al., 2018*). This obviously raises the question of the mechanism that prevents this silencing in attachment cells.

It is clear that we cannot exclude the possibility that an active mechanism, such as the SWI/SNF remodeling complexes, opens the chromatin structure in bi-fated cells to allow attachment cell dual behavior (*Hu et al., 2011*). However, such a mechanism cannot explain why the strategy of shared enhancers was selected. Finally, a mixed transcriptome is one of the hallmarks of stem cell pluripotency, as progenitor cells eventually chose one fate over the other upon differentiation (*Johnson et al., 2015*; *Soldatov et al., 2019*). Our findings suggest a different strategy, where using both programs facilitates the establishment of a specific attachment tissue. Overall, our results reveal a novel function for chromatin state, which allows the activation of two sets of genes in a third cell type to create a new cell fate that forms a transitional tissue.

KLF2 and KLF4 are known to regulate several biological processes, such as promoting the differentiation of gut and skin (KLF4, [*Katz et al., 2002*; *Segre et al., 1999*]) as well as the immune system (KLF2, [*Kuo et al., 1997*]), maintaining pluripotency of embryonic stem cells (KLF2 and KLF4, [*Jiang et al., 2008*]), and, together with other factors, inducing pluripotency to generate iPSC by reprogramming (KLF4, [*Takahashi and Yamanaka, 2006*]). Several works describe the involvement of KLF2 and KLF4 in the musculoskeletal system. In bones, *Klf4* over-expression in osteoblasts caused delayed bone development, in addition to impaired blood vessel invasion and osteoclast recruitment (*Michikami et al., 2012*). Another study showed that KLF2/4 are expressed during chick limb development in tendons and ligaments as part of the genetic program that regulates connective tissues (*Orgeur et al., 2018*). Previous studies showed that KLF2 and KLF4 display high similarity in protein sequences (*Dang et al., 2000*; *Shields and Yang, 1997*), suggesting that these factors could have common target sequences and may be functionally redundant. Indeed, loss of both KLF2 and KLF4 during embryogenesis led to abnormal blood vessel development and early lethality. This phenotype was more severe than what was observed in embryos that lost only KLF2 or KLF4 (*Chiplunkar et al., 2013*). Furthermore, previous work identified 313 target genes shared between KLF2 and KLf4, suggesting that they overlap in regulating gene expression (*Orgeur et al., 2018*). In this work, we show that KLF2/4 are central regulators of the attachment site. While the attachment did form initially in their absence, the subsequent differentiation failed, suggesting that KLF2/4 play a role at this stage. While we concentrated in this study on the attachment site, it is most likely that KLF2/4 play a role also in other musculoskeletal tissues such as the skeleton, tendon, and muscle. This possibility is supported by previous studies, where KLF2/4 were shown to be expressed in osteoblasts, chondrocytes, tenocytes, and muscle connective tissues (*Michikami et al., 2012*; *Orgeur et al., 2018*). Previous studies demonstrate the role of muscle-induced mechanical load in the development of attachment site (*Blitz et al., 2009*). In that context, our finding that KLF2/4 regulate the differentiation of attachment cells is interesting, because previous works have shown that these factors are mechanically regulated. It was shown in mice that shear stress on the vessels induced by blood flow leads to upregulation of *Klf2* expression (*Lee et al., 2006*). Additional in vitro studies showed that KLF2 and KLF4 are influenced by shear stress (*Chiplunkar et al., 2013*; *Dekker et al., 2005*; *Villarreal et al., 2010*). It is therefore possible that these factors are regulated by muscle forces, leading to the proper differentiation and maturation of the attachment site.

The ability of KLF2/4 to regulate gene expression in the attachment site is supported by our finding that many of the genes that were expressed by attachment cells had in their regulatory region KLF2/4 binding sites. Yet, it is clear that not all of them share this property, suggesting that these two factors are part of a larger transcriptional network. For example, our bioinformatic analysis identified other TF families such as GLI's, RUNX's, and NFI's as regulators of the attachment sites. *Gli1* was previously reported as a marker for enthesis cells (*Dyment et al., 2015*; *Felsenthal et al., 2018*; *Liu et al., 2012a*; *Schwartz et al., 2015*). Since gene expression by attachment cells is regulated by sharing enhancers with chondrocytes or tenocytes, it is reasonable to assume that some regulators of these cells might be part of the network that regulates the attachment cells. Indeed, loss of the tendon regulator *Scx* in mice led to failure of attachment cells to differentiate. Additionally, loss of

the chondrogenic regulator *Sox9* in *Scx*-expressing cells led to failure in attachment site formation (***Blitz et al., 2013***; ***Blitz et al., 2009***).

To conclude, by characterizing the transcriptome and chromatin landscape of tendon-to-bone attachment cells, we provide a molecular understanding of the bi-fated identity of these cells. Moreover, by identifying the transcription factors KLF2/4 as central regulators and the strategy of sharing enhancers with either tenocytes or chondrocytes, we provide a mechanism that regulates these bi-fated cells (***Figure 6***). These findings present a new concept for the formation of a border tissue, which is based on the simultaneous expression of a mixed transcriptome of the two flanking cell types by the intermediate cells. This strategy allows the formation of a unique transitional tissue without developing de novo a dedicated genetic program that regulates a third, new cell fate.

## Materials and methods

For Key Resources Table, see Appendix.

### Animals

The generation of floxed *Klf2* (***Lee et al., 2006***), floxed *Klf4* (***Katz et al., 2002***), *Prx1-Cre* (***Logan et al., 2002***), *Sox9-CreER* (***Soeda et al., 2010***), *Col2-CreER^T* (***Nakamura et al., 2006***), *Col2a1-Cre* (***Ovchinnikov et al., 2000***), *R26R-tdTomato* (***Madisen et al., 2010***), and *Scx-GFP* (***Pryce et al., 2007***) mice have been described previously.

To create *Col2a1-CreER, R26R-tdTomato, Sox9-CreER, R26R-tdTomato,* and *Col2a1-Cre, R26R-tdTomato* reporter mice, all on the background of *Scx-GFP*, floxed *R26R-tdTomato* mice were mated with *Col2a1-CreER, Sox9-CreER* or *Col2a1-Cre* mice, respectively. These strains were mated on a mixed background of C57BL/6 and B6.129 (ICR) mice and used for LCM and FACS experiments. To create *Prx1-Klf2-Klf4* mutant mice, floxed *Klf2-Klf4* mice were mated with *Prx1-Klf2-Klf4*

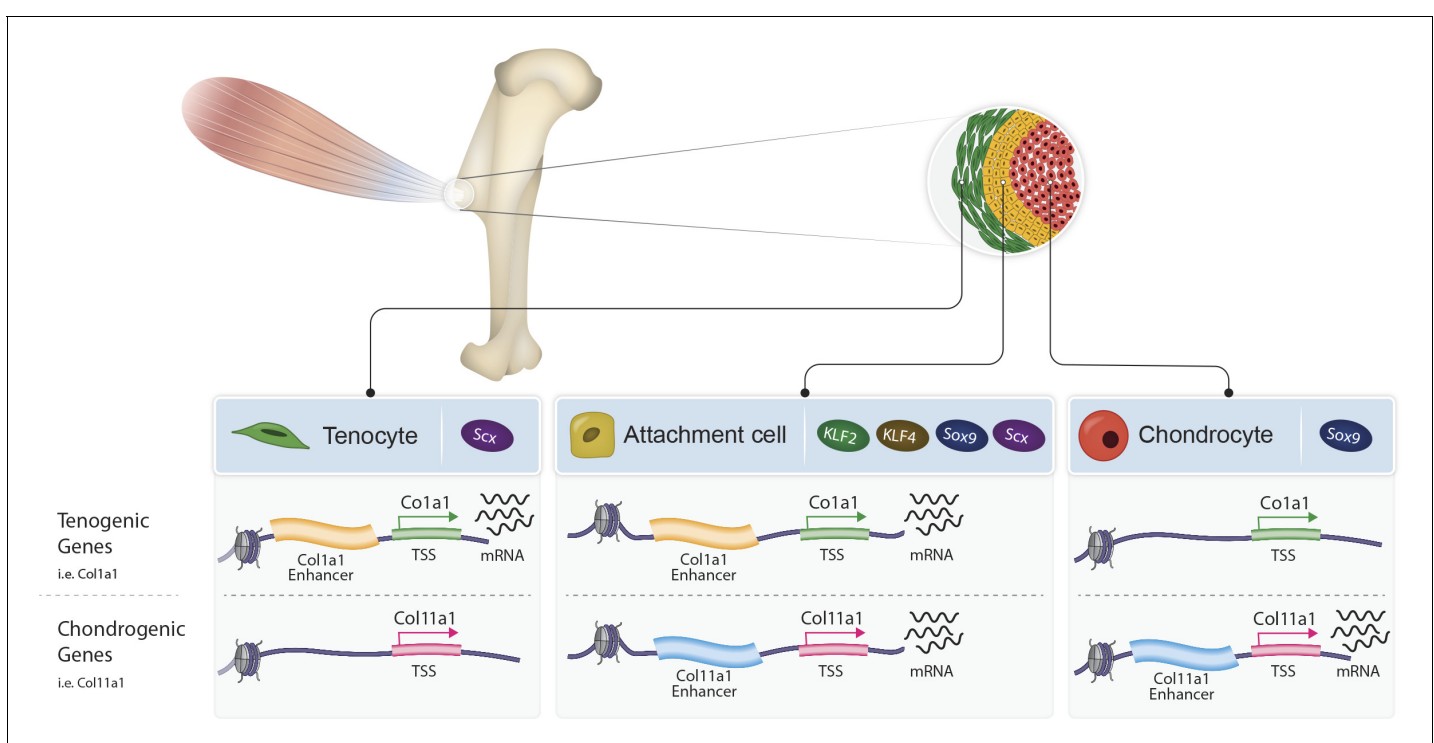

**Figure 6.** Proposed model of bi-fated tendon-to-bone attachment cells that are regulated by shared enhancers and KLF transcription factors. Tenocytes (green, left) and chondrocytes (red, right) express tenogenic (i.e. Col1a1) or chondrogenic (i.e. Col11a1) genes, respectively, whereas attachment cells (yellow, middle) express both chondrogenic and tenogenic genes to form the attachment site. Attachment cells duality of gene expression is regulated epigenetically by intergenic chromatin areas, which are accessible in these cells and in either tenocytes or chondrocytes. Additionally, at the transcriptional level, the transcription factors KLF2/4 are expressed by attachment cells and regulate their differentiation.

mice. As a control, we used embryos that lack *Cre* alleles. E14.5 wild-type C57BL/6 mice were used for ISH experiments as well.

For FACS experiments, *Sox9-CreER* or *Col2a1-CreER* mice were crossed with *Rosa26-tdTomato* reporter mice, all on the background of *Scx-GFP*. Induction of Cre recombinase was performed at various pregnancy stages by administration of 0.03 mg/gr tamoxifen/body weight in corn oil by oral gavage (stock concentration was 5 mg/ml). In all timed pregnancies, plug date was defined as E0.5. For harvesting of embryos, timed-pregnant females were sacrificed by cervical dislocation. Tail genomic DNA was used for genotyping by PCR.

## Laser capture microdissection

E14.5 *Col2a1-tdTomato-Scx-GFP* mouse forelimbs were dissected, shortly fixed in 4% PFA, washed with PBS and cryo-embedded (as described by *Bhattacherjee et al., 2004 Bhattacherjee et al., 2004*). Next, samples were cryo-sectioned and mounted on LCM slides (PET, Zeiss), washed with RNase-free water and EtOH (Arcturus dehydration component kit) according to an altered protocol of *Pazin et al., 2012*. LCM (PALM MicroBeam C system, Zeiss) was calibrated for refined tissue cutting. Isolated cells were collected to LCM caps (Adhesive Cap 500 clear, Zeiss) and RNA was purified using RNeasy FFPE Kit (Qiagen). The resulting RNA was the input for the bulk MARS-seq protocol (RNA sequencing).

## Preparation of single-cell suspension for fluorescence-activated cell sorting (FACS)

Flow cytometry analysis and sorting were performed at the Weizmann Institute of Science Flow Cytometry Core Facility on a BD FACS AriaIII instrument (BD Immunocytometry Systems) equipped with 488, 407, 561, and 633 nm lasers, using a 70 µm nozzle, controlled by BD FACS Diva software v8.0.1 (BD Biosciences). Further analysis was performed using FlowJo software v10.2 (Tree Star). For collection of cells, *Sox9-CreER^T2^-tdTomato;ScxGFP* or *Col2a1-CreER-tdTomato;ScxGFP* mice were crossed with *Rosa26-tdTomato;ScxGFP* reporter mice. Embryos were harvested at E13.5 following tamoxifen administration at E12.0, as described above. Forelimbs were dissected and suspended in cold PBS. To extract cells from tissues, PBS was replaced with 1 ml heated 0.05% trypsin and collagenase type V (dissolved in DMEM, Sigma) and incubated for 15 min at 37℃, gently agitated every 5 min. Tissues were then dissociated by vigorous pipetting using 1 ml tips. Next, 4 ml of DMEM supplemented with 10% FBS and 1% Pen-Strep was added and cell suspensions were filtered with 40 µm filter net. Finally, tubes were centrifuged at 1000 rpm for 7 min, supernatant was removed and cells were resuspended in 0.5–1 ml of cold PBS and used immediately for FACS. Single-stained GFP and tdTomato control cells were used for configuration and determining gate boundaries. Live cells were gated by size and granularity using FSC-A versus SSC-A and according to DAPI staining (1 µg/ml). FSC-W versus FSC-A was used to further distinguish single cells. In addition, unstained, GFP-stained only and tdTomato-stained only cells were mixed in various combinations to verify that the analysis excluded false-positive doublets. GFP was detected by excitation at 488 nm and collection of emission using 502 longpass (LP) and 530/30 bandpass (BP) filters. tdTomato was detected by excitation at 561 nm and collection of emission using a 582/15 BP filter. DAPI was detected by excitation at 407 nm and collection of emission using a 450/40 BP filter.

## Real-time PCR (RT-PCR)

Total RNA was purified from LCM-isolated samples of E14.5 mouse forelimbs using RNeasy FFPE Kit (Qiagen). Reverse transcription was performed with High Capacity Reverse Transcription Kit (Applied Biosystems) according to the manufacturer's protocol. Analysis of *Col2a1* and *Scx* was performed to monitor RNA quality during LCM calibrations, whereas RNA quantity was monitored by analysis of β–actin. RT-PCR was performed using Fast SYBR Green master mix (Applied Biosystems) on the StepOnePlus machine (Applied Biosystems). Values were calculated using the StepOne software. Data were normalized to *18S* rRNA or β-actin in all cases.

## In situ hybridization

Section ISH were performed as described previously (*Riddle et al., 1993*). Single-and double-fluorescent ISH on paraffin sections were performed using DIG- and/or FITC-labeled probes (listed in

**Table 4.** List of probes used for in situ hybridization.

| Probe name | Genomic position | Refseq template | Size (bp) |
|---|---|---|---|
| Igfbp5 | 792–1264 | NM_010518.2 | 502 |
| Col11a1 | 831–1224 | NM_007729.3 | 393 |
| Biglycan | 147–716 | NM_007542.4 | 569 |
| Wwp2 | 2488–3056 | NM_025830.3 | 568 |
| Col5a1 | 626–1298 | NM_015734.2 | 672 |
| Col1a1 | 4295–4475 | NM_007742.4 | 180 |
| Klf2 | 895–1512 | NM_008452.2 | 617 |
| Gli1 | 1810–2447 | NM_010296.2 | 638 |
| Bsp | 145–1058 | NM_008318.3 | 1955 |
| Scx | 273–1129 | NM_198885.3 | 856 |
| Mfap4 | 276–966 | NM_029568.2 | 690 |
| Ptn | 1247–1932 | NM_008973.2 | 685 |
| Col3a1 | 707–1388 | NM_009930.2 | 681 |
| Mgp | 69–553 | NM_008597.4 | 485 |
| Eln | 154–693 | NM_007925.4 | 540 |
| Mmp14 | 708–1283 | NM_008608.3 | 575 |
| Col2a1 | 4474–4879 | NM_001113515.2 | 406 |
| Col9a1 | 2553–3100 | NM_007740.3 | 547 |
| Sox9 | 50–797 | NM_011448.4 | 748 |
| Snorc | 32–437 | NM_028473.1 | 405 |

*Table 4*; *Shwartz and Zelzer, 2014*). After hybridization, slides were washed, quenched, and blocked. Probes were detected by incubation with anti-DIG-POD (Roche; 1:300) and anti-FITC-POD (Roche, 1:200), followed by Cy2-tyramide- and Cy3-tyramide-labeled fluorescent dyes according to the instructions of the TSA Plus Fluorescent Systems Kit (Perkin Elmer).

## Single-molecule fluorescent in situ hybridization (smFISH)

Harvested E14.5 forelimbs were fixed with cold 4% formaldehyde (FA) in PBS and incubated first in 4% FA/PBS for 3 hr, then in 30% sucrose in 4% FA/PBS overnight at 4°C with constant agitation. Fixed tissues were embedded in OCT and sectioned at a thickness of 10 µm. The preparation of the probe library, hybridization procedure, and imaging conditions were previously described (*Itzkovitz et al., 2012*; *Lyubimova et al., 2013*; *Raj et al., 2008*). In brief, probe libraries were designed against biglycan (*Bgn*) and *Wwp2* mRNA sequences using the Stellaris FISH Probe Designer (Biosearch Technologies, Inc, Petaluma, CA) coupled to Quasar 670 and CAL Fluor Red 610, respectively. Libraries consisted of 17–96 probes each of length 20 bps, complementary to the coding sequence of each gene (*Supplementary file 2*). Nuclei were stained with DAPI. To detect cell borders, Alexa Fluor 488 conjugated phalloidin (Thermo Fisher, A12379) was added to the GLOX buffer, which was wash for 15 min. Slides were mounted using ProLong Gold (Molecular Probes, P36934).

## Image acquisition and analysis

For smFISH image acquisition, we used a Nikon-Ti-E inverted fluorescence microscope equipped with a Photometrics Pixis 1024 CCD camera to image 10-µm-thick cryosections. For image analysis, we used ImageM, a custom MATLAB program (*Lyubimova et al., 2013*), which was used to compute single-cell mRNA concentrations by segmenting each cell manually according to the cell borders and the nucleus. The size of the nucleus was detected automatically by the program according to the DAPI signal. For each cell, the concentration of cytoplasmic mRNA of each gene was

calculated by measuring the number of dots per volume. Images were visualized and processed using ImageJ 1.51 hr (*Schindelin et al., 2012*) and Adobe Illustrator CC2018.

## Bulk RNA sequencing

For this analysis, we adapted the MARS-seq protocol (*Jaitin et al., 2014*; *Keren-Shaul et al., 2019*) to generate RNA-seq libraries for expression profiling of the purified RNA from E14.5 LCM-isolated samples. Briefly, RNA from each sample was barcoded during reverse transcription and pooled. Following Agencourt Ampure XP beads cleanup (Beckman Coulter), the pooled samples underwent second strand synthesis and were linearly amplified by T7 in vitro transcription. The resulting RNA was fragmented and converted into a sequencing-ready library by tagging the samples with Illumina sequences during ligation, RT, and PCR. Libraries were quantified by Qubit and TapeStation as well as by qPCR for actb gene as previously described (*Jaitin et al., 2014*; *Keren-Shaul et al., 2019*). Sequencing was done on a Hiseq 2500 SR50 cycles kit (Illumina).

The data were analyzed using the Pipeline Pilot-designed pipeline for transSeq (by INCPM, https://incpmpm.atlassian.net/wiki/spaces/PUB/pages/36405284/tranSeq+on+Pipeline-Pilot). Briefly, the analysis included adapter trimming, mapping to the mm9 genome, collapsing of reads with the same unique molecular identifiers (UMI) of 4 bases (R2) and counting of the number of reads per gene with HTseq-count (*Anders et al., 2015*), using the most 3′ 1000 bp of each RefSeq transcript. DESeq2 (version 1.4.5, *Love et al., 2014*) was used for normalization and differential expression analysis with betaPrior set to true, cooksCutoff=FALSE, independentFiltering=FALSE. Benjamini-Hochberg method was used to adjust the raw p-values for multiple testing. Genes with adjusted p-value$\leq$0.05 and fold change $\geq$ 2 between every two conditions were considered as differential. PCA analysis was done using log-transformed normalized data (DESeq2 function rlogTransformation with parameter blind=TRUE) with a modified plotPCA function of DESeq2. Clustering of the log-normalized read count of differentially expressed genes was done using CLICK algorithm (Expander package version 7.1, *Ulitsky et al., 2010*), followed by visualization by R (*R Development Core Team, 2013*). Further analysis was performed using GSEA (Broad institute) and Gorilla (*Eden et al., 2009*; *Subramanian et al., 2005*). The crude data of the work has been deposited on NCBI GEO (GSE144306).

## Single-cell library preparation using chromium 10x genomics platform

Cells were counted and diluted to a final concentration in PBS supplemented with 0.04% BSA. Cellular suspension was loaded onto Next GEM Chip G targeting 4000 cells and then ran on a Chromium Controller instrument to generate GEM emulsion (10x Genomics). Single-cell 3′ RNA-seq libraries were generated according to the manufacturer's protocol (10x Genomics Chromium Single Cell 3′ Reagent Kit User Guide v3/v3.1 Chemistry).

## Next-generation sequencing of single-cell libraries

Single-cell 3′ RNA-seq libraries were quantified using NEBNext Library Quant Kit for Illumina (NEB) and high sensitivity D1000 TapeStation (Agilent). Libraries were pooled according to targeted cell number, aiming for ~50,000,000 reads per cell. Pooled libraries were sequenced on a NovaSeq 6000 instrument using an SP 100 cycles reagent kit (Illumina) (R1 28 bases, R2 82 bases, and I1 eight bases), specifically 290M fragments of the relevant library were sequenced.

## scRNA-seq bioinformatic analysis

Demultiplexing and alignment were performed using cellranger (10x Genomics version 3.0.2) bioinformatics pipeline using mm10 genome, followed by detecting swapped barcodes between libraries, since the NovaSeq run contained a mix of three libraries. For this, we used R (3.6.3) and the package DropletUtiles (*Lun et al., 2019*) (1.6.1) with the function swappedDrops (parameter min.frac = 0.9) and the molecule_info.h5 input. 1.5% of the molecules were detected as swapped. Detecting empty droplets was done with the function emptyDrops (parameter lower = 300), the number of cells detected was 2213.

For additional analysis, the R package Seurat (3.1.5) was used (*Butler et al., 2018*). The analysis included filtering genes (must be expressed in at least three cells), filtering cells with over 20% expression of mitochondria genes and high and low 3% percentiles of total number of genes and

UMIs; this resulted in 1218 cells. In addition, one of the clusters had significantly lower RNA counts as compare to other collected cells, which might have indicated that it contained mainly low-quality cells. Indeed, examination of the 10 most highly expressed genes using heatmap demonstrated this; therefore, this cluster was removed from further analysis. A total of 1076 cells were clustered using 2000 variable genes and 15 principal components (PCs) (resolution = 0.4). UMAP plot was made with the Seurat functions RunUMAP and DimPlot and violin plots were made with the VpnPlot function. The crude data of the work have been deposited on NCBI GEO (GSE160090).

## Assay for transposase-accessible chromatin with high-throughput sequencing

ATAC-seq data were trimmed from their adaptors and filtered from low quality reads using Cutadapt followed by alignment to the mm10 genome (GRCm38.p5) using Bowtie2 (version 2.3.4.1) (*Langdon, 2015*). PCR-duplicate reads were removed with Picard 'MarkDuplicates' (http://broadinstitute.github.io/picard/). Mitochondrial reads were removed from the alignment, and the data were further filtered to contain only reads with a unique mapping with SAMtools (-F 4 f 0 × 2). Read pairs with inner distance of up to 120 bp were selected as representing the accessible chromatin region. MACS2 (version 2.1.1.20160309) (*Zhang et al., 2008*) was applied for peak calling using the setting: callpeak -f BAMPE–nomodel. Peaks from all samples were combined and merged with BEDTools (*Quinlan, 2014*), followed by extension to a minimum length of 500 bp. For every tissue, a set of reproducible peaks was obtained by voting, which means that a normalized read count $\geq 30$ was detected in at least 50% of the replicates. Peaks that were not reproducible in any tissue were removed. Peaks that reside in the ENCODE 'Blacklist' regions, that is regions that were previously found by ENCODE (*ENCODE Project Consortium, 2012*) to produce artificial signal (http://mitra.stanford.edu/kundaje/akundaje/release/blacklists/), were also eliminated. Peak quantification was done with BedTools *Quinlan, 2014* following by DESeq2 (*Love et al., 2014*) normalization. Peaks with an averaged normalized read count $\geq 30$ in at least one of the studied tissues were selected for the downstream analyses. The crude data of the work have been deposited on NCBI GEO (GSE144306).

## Annotation and genomic feature enrichment analysis

Annotation of ATAC-seq peaks was performed using HOMER (*Heinz et al., 2010*) and GREAT (*McLean et al., 2010*). When a peak was associated by GREAT to multiple genes, the two closest genes were selected for further analysis. ATAC-seq peaks that were at a distance of up to −2 kb down or +0.5 kb up from a TSS of their annotated gene (HOMER) were considered as promoter peaks; otherwise, peaks were considered as distal. To rank distal ATAC-seq peaks as putative cis-regulatory elements, we calculated the overlap between the peaks and relevant histone modification datasets (ChIP-seq) performed by the ENCODE project (*Yue et al., 2014*) on E13.5 C57BL/6 mouse embryo limb. The overlap was calculated using BEDTools intersect (*Quinlan, 2014*). The following datasets were used: ENCSR905FFU (H3K27ac) and ENCSR426EZM (H3K4me1) as markers of enhancers, ENCSR416OYH (H3K4me3) as a marker of promoters and ENCSR022DED (H3K9me3). Overlap with the phastConsElements60wayPlacental track downloaded from the UCSC site (*Casper et al., 2018*) was calculated to account for evolutionary conservation. Enrichment analysis of over-representation of TFBSs in the ATAC-seq peaks was performed with the RegionMiner tool of Genomatix and HOMER.

## Enhancer reporter assays in mouse embryos

Candidate enhancers were PCR-amplified and cloned upstream of a *Shh*-promoter-LacZ-reporter cassette. We used a mouse enhancer-reporter assay that relies on site-specific integration of a transgene into the mouse genome (*Kvon et al., 2020*). In this assay, the reporter cassette is flanked by homology arms targeting the safe harbor locus (*Tasic et al., 2011*). Cas9 protein and a sgRNA targeting H11 were co-injected into the pronucleus of FVB single cell-stage mouse embryos (E0.5) together with the reporter vector (*Kvon et al., 2020*). Embryos were sampled and stained at E14.5. Embryos were excluded from further analysis if they did not carry the reporter transgene. All mouse works were reviewed and approved by the Lawrence Berkeley National Laboratory Animal Welfare and Research Committee.

## Acknowledgements

We thank Nitzan Konstantin for expert editorial assistance, Dr. Douglas Lutz and service engineer Tal Alon (Getter Bio-Med, Zeiss) for LCM calibration, Neria Sharabi from the Department of Veterinary Resources, Weizmann Institute, and all Zelzer lab members for suggestions and advice. We thank Drs. Merav Kedmi, David Pilzer and Hadas Keren-Shaul from the Genomics Sandbox unit at the Life Science Core Facility, Weizmann Institute of Science, for critical advice. We thank E Sebzda for providing floxed *Klf2* mice and the Mutant Mouse Regional Resource Center (MMRRC) at UC Davis for providing floxed *Klf4* mice. We thank the ENCODE Consortium and the ENCODE production laboratory for generating the described datasets. This study was supported by grants from the Israel Science Foundation (ISF, grant No. 345/16), National Institute of Health (grant No. R01 AR055580), Minerva Foundation (grant No. 713533), the Estate of Mr. and Mrs. van Adelsbergen, and the David and Fela Shapell Family Center for Genetic Disorders (to EZ). Work at Lawrence Berkeley National Lab was supported by National Institute of Health grant R01HG003988 (to AV) and performed under Department of Energy Contract DE-AC02-05CH11231, University of California. MO was supported by Swiss National Science Foundation grant PCEFP3_186993.

## Additional information

### Funding

| Funder | Grant reference number | Author |
|---|---|---|
| National Institutes of Health | R01 AR055580 | Elazar Zelzer |
| National Institutes of Health | R01HG003988 | Axel Visel |
| Israel Science Foundation | grant No. 345/16 | Elazar Zelzer |
| Minerva Foundation | grant No. 713533 | Elazar Zelzer |
| David and Fela Shapell Family Center for Genetic Disorders | | Elazar Zelzer |
| Estate of Mr. and Mrs. van Adelsberge | | Elazar Zelzer |
| University of California | DEAC02-05CH11231 | Marco Osterwalder |
| Swiss National Science Foundation | PCEFP3_186993 | Marco Osterwalder |

The funders had no role in study design, data collection and interpretation, or the decision to submit the work for publication.

### Author contributions

Shiri Kult, Conceptualization, Validation, Investigation, Visualization, Methodology, Writing - original draft; Tsviya Olender, Dena Leshkowitz, Data curation, Software, Formal analysis, Validation, Visualization, Methodology, Writing - review and editing; Marco Osterwalder, Validation, Investigation, Visualization, Methodology, Writing - review and editing; Svetalana Markman, Visualization, Writing - review and editing; Sharon Krief, Validation, Investigation, Visualization, Methodology; Ronnie Blecher-Gonen, Investigation, Methodology; Shani Ben-Moshe, Investigation, Visualization, Methodology, Writing - review and editing; Lydia Farack, Software, Methodology, Writing - review and editing; Hadas Keren-Shaul, Investigation, Methodology, Writing - review and editing; Tomer-Meir Salame, Data curation, Formal analysis, Validation, Methodology, Writing - review and editing; Terence D Capellini, Resources, Writing - review and editing; Shalev Itzkovitz, Resources, Methodology, Writing - review and editing; Ido Amit, Resources, Methodology; Axel Visel, Resources, Funding acquisition, Methodology, Writing - review and editing; Elazar Zelzer, Conceptualization, Supervision, Funding acquisition, Methodology, Writing - original draft, Project administration

### Author ORCIDs

Shiri Kult http://orcid.org/0000-0001-8216-2908
Marco Osterwalder http://orcid.org/0000-0002-1969-2313

Lydia Farack [iD] http://orcid.org/0000-0001-9597-5078
Terence D Capellini [iD] http://orcid.org/0000-0003-3842-8478
Shalev Itzkovitz [iD] http://orcid.org/0000-0003-0685-2522
Axel Visel [iD] http://orcid.org/0000-0002-4130-7784
Elazar Zelzer [iD] https://orcid.org/0000-0002-1584-6602

## Ethics

Animal experimentation: All mice were maintained and used in accordance with protocols approved by the Weizmann Institutional Animal Care and Use Committee (IACUC; permission no. 10550119-2). All animal work was reviewed and approved by the Lawrence Berkeley National Laboratory (LBNL) Animal Welfare Committee. All mice used in this study were housed at the Animal Care Facility (ACF) at LBNL. Mice were monitored daily for food and water intake, and animals were inspected weekly by the Chair of the Animal Welfare and Research Committee and the head of the animal facility in consultation with the veterinary staff. The LBNL ACF is accredited by the American Association for the Accreditation of Laboratory Animal Care (AAALAC). Transgenic mouse assays were performed in *Mus musculus* FVB background mice.

## Decision letter and Author response

Decision letter https://doi.org/10.7554/eLife.55361.sa1
Author response https://doi.org/10.7554/eLife.55361.sa2

## Additional files

### Supplementary files

• Supplementary file 1. Hierarchical clustering of bulk MARS-seq data from E14.5 attachment site samples. Ordered genes and samples were combined by complete-linkage clustering using the similarity measurement of Pearson correlation. A, remote tenocytes; B, adjacent tenocytes; C, remote chondrocytes; D, adjacent chondrocytes; E, attachment cells.

• Supplementary file 2. List of probes used for single-molecule fluorescent in situ hybridization for biglycan (*Bgn*) and *Wwp2.*

• Supplementary file 3. List of intergenic regions (non-TSS). The peaks were selected based on their enhanced ATAC-seq signal (cutoff of averaged ATAC-seq signal > 30.0) and their association with differentially expressed genes. 'Attachment specific' sheet lists 248 peaks that are accessible only in attachment cells. The table includes transcription factors such as *Klf4*, *Runx3*, and *Nfia*, in addition to ECM-associated genes such as *Col11a1* and *Bgn*.

• Transparent reporting form

### Data availability

The data was submitted to GEO (GSE144306). Additional data of scRNA-seq was submitted to GEO (GSE160090).

The following datasets were generated:

| Author(s) | Year | Dataset title | Dataset URL | Database and Identifier |
|---|---|---|---|---|
| Kult S, Olender T, Osterwalder M, Markman S, Leshkowitz D, Krief S, Blecher-Gonen R, Ben-Moshe S, Farack L, Keren-Shaul H, Meir Salame T, Capellini TD, Itzkovitz S, Amit I, Visel A, Zelzer E | 2020 | Bi-fated tendon-to-bone attachment cells are regulated by shared enhancers and KLF transcription factors | https://www.ncbi.nlm.nih.gov/geo/query/acc.cgi?acc=GSE144306 | NCBI Gene Expression Omnibus, GSE144306 |

| Kult S, Olender T, Osterwalder M, Markman S, Leshkowitz D, Krief S, Blecher-Gonen R, Ben-Moshe S, Farack L, Keren-Shaul H, Meir Salame T, Capellini TD, Itzkovitz S, Amit I, Visel A, Zelzer E | 2020 | Bi-fated tendon-to-bone attachment cells are regulated by shared enhancers and KLF transcription factors | https://www.ncbi.nlm.nih.gov/geo/query/acc.cgi?acc=GSE160090 | NCBI Gene Expression Omnibus, GSE160090 |

The following previously published datasets were used:

| Author(s) | Year | Dataset title | Dataset URL | Database and Identifier |
|---|---|---|---|---|
| Ren B, UCSD | 2015 | H3K27ac ChIP-seq on embryonic 13.5 day mouse limb | https://www.encodeproject.org/experiments/ENCSR905FFU/ | ENCODE, ENCSR905FFU |
| Ren B, UCSD | 2015 | H3K4me1 ChIP-seq on embryonic 13.5 day mouse limb | https://www.encodeproject.org/experiments/ENCSR426EZM/ | ENCODE, ENCSR426EZM |
| Ren B, UCSD | 2015 | H3K4me3 ChIP-seq on embryonic 13.5 day mouse limb | https://www.encodeproject.org/experiments/ENCSR416OYH/ | ENCODE, ENCSR416OYH |
| Ren B, UCSD | 2015 | H3K9me3 ChIP-seq on embryonic 13.5 day mouse limb | https://www.encodeproject.org/experiments/ENCSR022DED/ | ENCODE, ENCSR022DED |

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

# Appendix 1

**Appendix 1—key resources table**

| Reagent type (species) or resource | Designation | Source or reference | Identifiers | Additional information |
|---|---|---|---|---|
| Strain, strain background (*Mus musculus*) | *Sox9-CreER*, C57BL/6 | DOI:10.1002/dvg.20667 | | FACS |
| Strain, strain background (*M. musculus*) | *R26R-tdTomato*, C57BL/6 | DOI:10.1038/nn.2467 | | FACS |
| Strain, strain background (*M. musculus*) | *Scx-GFP*, C57BL/6 | DOI:10.1002/dvdy.21179 | RRID:MGI:3717422 | FACS |
| Strain, strain background (*M. musculus*) | *Col2a1-Cre*, C57BL/6 and B6.129 (ICR) | PMID:10686612 | | Used as mixed background (C57BL/6 and B6.129 (ICR)) for LCM |
| Strain, strain background (*M. musculus*) | *Col2-CreER$^T$*, C57BL/6 and B6.129 (ICR) | DOI:10.1002/dvdy.20892 | RRID:IMSR_JAX:006774 | Used as mixed background (C57BL/6 and B6.129 (ICR)) for FACS |
| Strain, strain background (*M. musculus*) | Floxed *Klf2*, C57BL/6 | Eric Sebzda | | DOI:10.1016/j.devcel.2006.09.006 |
| Strain, strain background (*M. musculus*) | Floxed *Klf4*, C57BL/6 | MMRRC* | RRID:MMRRC_029877-MU | PMID:12015290 *Mutant Mouse Regional Resource Center at UC Davis |
| Strain, strain background (*M. musculus*) | *Prx1-Cre*, C57BL/6 | DOI:10.1002/gene.10092 | | |
| Strain, strain background (*M. musculus*) | C57BL/6 | The Jackson Laboratory | | In situ hybridization and single-molecule fluorescent in situ hybridization |
| Strain, strain background (*M. musculus*) | FVB/Shh-ZRSem7Axvi (396C>T variant knock-in), FVB | DOI:10.1016/j.cell.2020.02.031 | | Enhancer reporter assays in mouse embryos |
| Strain, strain background (*M. musculus*) | FVB | Charles River | https://www.criver.com/ | Enhancer reporter assays in mouse embryos |
| Antibody | anti-DIG-POD (sheep polyclonal) | Roche | Cat# 11207733910 | ISH (1:300), DOI:10.1007/978-1-62703-989-5_15 |
| Antibody | anti-FITC-POD (sheep polyclonal) | Roche | Cat# 11426346910 | ISH (1:200), DOI:10.1007/978-1-62703-989-5_15 |

*Continued on next page*

*Appendix 1—key resources table continued*

| Reagent type (species) or resource | Designation | Source or reference | Identifiers | Additional information |
|---|---|---|---|---|
| Commercial assay or kit | TSA Plus Fluorescent Systems Kit | Perkin Elmer | Cat# NEL753001KT | ISH (Cy3 + fluorescein), DOI:10.1007/978-1-62703-989-5_15 |
| Antibody | Anti-SOX9 (rabbit polyclonal) | Millipore | AB5535 | (1:200) |
| Commercial assay or kit | Histogene LCM Frozen Section Staining Kit | ThermoFisher Scientific | Cat# KIT0401 | LCM |
| Commercial assay or kit | MembraneSlide 1.0 PET | Carl Zeiss Microscopy | Cat# 415190-9051-000 | LCM |
| Commercial assay or kit | AdhesiveCap 500 clear | Carl Zeiss Microscopy | Cat# 415190-9211-000 | LCM |
| Commercial assay or kit | RNeasy FFPE Kit | Qiagen | Cat# 73504 | LCM |
| Sequence-based reagent | smFISH probes | Stellaris FISH Probe Designer (Biosearch Technologies, Inc, Petaluma, CA) | | See *Supplementary file 2* |
| Software, algorithm | Pipeline Pilot-designed pipeline for transSeq | INCPM | https://incpmpm.atlassian.net/wiki/spaces/PUB/pages/36405284/tranSeq+on+Pipeline-Pilot | Bulk RNA sequencing |
| Software, algorithm | HTseq-count | DOI: 10.1093/bioinformatics/btu638 | | |
| Software, algorithm | DESeq2 | DOI:10.1186/s13059-014-0550-8 | Version 1.4.5 | |
| Software, algorithm | Expander package | DOI: 10.1038/nprot.2009.230 | Version 7.1, CLICK algorithm | |
| Software, algorithm | cellranger | 10x Genomics | Version 3.0.2 | scRNA-seq bioinformatic analysis |
| Software, algorithm | DropletUtiles | DOI:10.1186/s13059-019-1662-y | Version 1.6.1 | scRNA-seq bioinformatic analysis |
| Software, algorithm | Seurat | DOI:10.1038/nbt.4096 | Version 3.1.5 | scRNA-seq bioinformatic analysis |
| Other | Bulk RNA sequencing dataset | This paper | NCBI GEO (GSE144306) | |
| Other | Single-cell RNA sequencing dataset | This paper | NCBI GEO (GSE160090) | |
| Commercial assay or kit | Hiseq 2500 SR50 cycles kit | Illumina | | Bulk RNA sequencing |
| Commercial assay or kit | 10x Genomics Chromium Single Cell 3' Reagent Kit | 10x Genomics | | User Guide v3/v3.1 Chemistry |
| Commercial assay or kit | NEBNext Library Quant Kit for Illumina | NEB | | |
| Commercial assay or kit | SP 100 cycles reagent kit | Illumina | | |
| Other | Alexa Fluor 488 conjugated phalloidin | ThermoFisher Scientific | A12379 | smFISH |
| Other | ProLong Gold | Molecular Probes | P36934 | smFISH |

*Continued on next page*

*Appendix 1—key resources table continued*

| Reagent type (species) or resource | Designation | Source or reference | Identifiers | Additional information |
|---|---|---|---|---|
| Commercial assay or kit | NextSeq 500 High Output v2 Kit (75 cycles) | Illumina | FC-404–2005 | |
| Commercial assay or kit | Nextera Index Kit 24ind, 96smp | Illumina | FC-121–1011 | |
| Commercial assay or kit | Nextera DNA Sample Prep Kit (24 sam) | Illumina | FC-121–1030 | |
| Software, algorithm | Bowtie2 | DOI:10.1186/s13040-014-0034-0 | Version 2.3.4.1 | |
| Software, algorithm | Picard 'MarkDuplicates' | http://broadinstitute.github.io/picard/ | Version 1.119 | |
| Software, algorithm | MACS2 | DOI:10.1186/gb-2008-9-9-r137 | Version 2.1.1.20160309 | |
| Software, algorithm | BEDtools | DOI:10.1002/0471250953.bi1112s47 | Version 2.27.1 | |
| Software, algorithm | SAMtools | DOI:10.1093/bioinformatics/btp352 | Version 1.19 | |
| Software, algorithm | GREAT | DOI: 10.1038/nbt.1630 | Version 4.0.4 | |
| Software, algorithm | HOMER | PMID:20513432 | Version 1.9 | |
| Software, algorithm | Genomatix | | | |
| Other | ATAC-seq dataset | This paper | NCBI GEO (GSE144306) | |
| Other | histone modification datasets H3K27ac, H3K4me1, H3K4me3, H3K9me3 | ENCODE Consortium and the ENCODE production laboratory | ENCSR905FFU ENCSR426EZM ENCSR416OYH ENCSR022DED | ChIP-seq |
| Other | phastConsElements60wayPlacental | UCSC site; *Casper et al., 2018* | | |

