## [Decision Letter]

**Acceptance summary:**

Your study describes bi-fated cells at the murine tendon-to-bone attachment, namely with activation of a mixture of chondrocyte and tenocyte transcriptomes, under the regulation of shared regulatory elements and KLF transcription factors, notably KLF2 and KLF4. The report provides novel insights into the yet unknown molecular and cellular architecture of the tendon-to-bone attachments, and hence is seen to be both novel and medically relevant.

**Decision letter after peer review:**

Thank you for submitting your article "Bi-fated tendon-to-bone attachment cells are regulated by shared enhancers and KLF transcription factors" for consideration by *eLife*. Your article has been reviewed by three peer reviewers, one of whom is a member of our Board of Reviewing Editors, and the evaluation has been overseen by Clifford Rosen as the Senior Editor. The reviewers have opted to remain anonymous.

The reviewers have discussed the reviews with one another and the Senior Editor has drafted this decision to help you prepare a revised submission.

The manuscript from Kult and colleagues focuses on the transcriptional regulation of a unique set of cells located between bone and tendon known as the attachment unit. Using RNA-seq and ATAC-seq, the authors investigate shared and unique transcriptional programs and accessible genomic regions. The ATACseq and epigenome profiling reveals transcriptional enhancers, with overlapping intergenic areas between bifated and both fates. Transgenic enhancer reporters expose common enhancers for tenocytes and chondrocytes. *Klf2* and *Klf4* were identified as critically required for differentiation. These findings lead the authors to propose that bi-fated attachment cells that connect tendon to bone. Previous studies showed that these cells express cartilage and tendon markers so the present study would need to clearly highlight the advances made compared to previous work. As a general note for example, the co-option of existing enhancers does not rule out the existence of de novo ones. This needs to be addressed clearly in the study.

Essential revisions:

1) The test of enhancers by transgenesis is somewhat limited in scope (only 8 tested) and yields some surprising results that can be explored further. An important request would be to make sure of the reproducibility of experiments. It is said that 5 out of 8 selected elements drove expression in the forelimb, but not how many attempts were done for the negative ones. Moreover, as reported in the transparent report form, N=1 for *Col1a1* element, N=3 for *Klf2* element, N=2 for *Sox9* element, N=3 for Mgp element, N=4 for *Col11a1*. The reviewers understand that a site-directed approach is likely to be more reproducible that random insertion, it is recommended to examine at least 3 instances per element, to be certain of some surprising results. For example, although *Sox9* is not expressed in hypertrophic chondrocytes, the *Sox9* element drives expression in hypertrophic chondrocytes. Moreover, *Sox9* and *Klf2* elements drive expression in hypertrophic chondrocytes but not in AU cells. If these results are confirmed, they may cast doubt on the conclusion that co-option of enhancers is the (only) mechanism that regulates expression in AU cells.

2) The authors have not systemically looked for AU enhancers that are not shared with tenocytes or chondrocytes. Combined ATAC-seq dataset and published ChIP-seq of histone marks can potentially identify new enhancers. The authors could speculate or assess if those enhancers were acquired de novo and exclusive of AU cells.

3) Figure 1A, the authors present a PCA biplot : Can they be more specific on how the data were transformed prior the dimension reduction (FPKM, VST, Log transformed, CPM…?). How many genes were taken into account in the PCA; All or as it is more commonly done on the 500 most variant ones ?

4) Figure 1 C – GO terms found are very generic, this information does not really seem to be useful. Can the authors can be more specific on the parameters they used in their GSEA analysis : test used and p-value correction (FDR q-value suggests a Benjamin and Hochberg correction, it that right ?

5) In general, one has to give the detail on the software version (including package version) and OS type used for the bioinformatic analysis, these informations are missing from the manuscript.

6) In the sentence: "This suggests that the attachment cell transcriptome is largely shared with both chondrocytes and tenocytes (Figure 1A, PC1 52.47%)", the word largely is misleading.

7) The authors do not discuss the variation both on the first and the second PC and of the attachment samples. This is a big issue because there are only 2 samples for this category of cells in which the intra-group variability is very high. This leads to a poor statistical parameter estimation giving rise to poor statistical test outcome.

8) Legend of Figure 1: The term MARS-Seq is slightly misleading as it is usually associated with single cell RNAseq analysis. For clarity, please write instead bulk-MARS-Seq.

9) "To further support our initial observation that the transcriptome of the attachment cells is a mixture of chondrocyte and tenocyte transcriptomes, we clustered the statistically significant differentially expressed genes between all samples into 5 clusters, using CLICK". Same remark as the use of contrasts in DESeq2.

10) In the sentence : "From these two clusters, 374 genes, 320 of them tenogenic and 54 chondrogenic, were also found to be expressed by attachment cells." It is unclear what "so found to be expressed by attachment cells" mean? For instance, for the tenogenic markers, does this mean that in the attachment vs. chondrocytes comparison these genes are up-regulated in the attachment cells? In that case, how are the tenogenic markers defined, using the tenocytes vs chondrocytes comparison? Would it be possible to have a Venn diagram to help follow the process to define the different marker identifications? Has a simpler method using contrasts in DESeq2 been tested? If yes, do the results converge with the ones presented here? Are these "statistically significant differentially expressed genes between all samples" coming from a pairwise wald-test or a likelihood ratio test? the Materials and methods suggest that the wald-test was used. Please clarify.

11) In a previous study the authors investigated the emergence of the attachment unit (AU) with focus on bone eminence progenitors (co-expressing *Sox9* and SCX up to E12.5 and expressed Col2 after E12.5 according to Col2a1CreERT2 lineage tracing). Here, they focus on the transcriptome of E14.5 attachment cells from the deltoid tuberosity, however these appear to be different from tuberosity progenitors (adjacent chondrocytes) as described in Figure 1—figure supplement 1. A better definition of what is defined as attachment unit in this paper vs previous papers and/or the AU subcompartments would help clarify the populations that are being examined.

12) Moreover, as the constitutive Col2a1Cre did not label the AU, but in the previous study did label the AU/bone eminence progenitors, it is unclear what the exact definition of AU is.

13) When using the Col2a1-Cre, R26R-tdTomato and Scx-GFP, the authors mention : Unexpectedly, the two reporters failed to label the attachment cells that were located in between these two populations. This failure might be due to a missing regulatory element in one of the constructs that was used to produce each transgenic reporter. However, in Figure 1—figure supplement 1, subpopulation 5 seems to have SCX+ cells. Is this an error of labelling? What is the orientation of this section?

14) For FACS and ATACseq analysis the authors use Sox9CreERT2;tdTomato;SCX-GFP and Col2CreERT2;tdTomato;SCX-GFP. It is not clear why for FACS the Col2CREERT2 line is used while for LCM the constitutive one is used. Moreover, as previously reported, they isolate attachment cells as double positive *SOX9*/SCX cells. Here again, do the cells taken for analysis include those of the tuberosity itself? Col2CreERT2 with Tamox at E12.5 should be labeling the tuberosity too. It is unclear which cells from which Cre/reporter combination have been used for the ATACseq experiments of Figure 3.

15) For Figure 2, a scheme showing where exactly in the bone we are located and how it has been sectioned would be helpful. Also, it would be nice to perform single molecule FISH on top of Col2Cre:R26TOM:SCX lineage tracing to show the specificity of the colocalization in the "double reporter-negative" area. Also, including the KLF2/4 FISH at this point would help visualize distinctions between genes belonging to cluster 5 (unique to AU) vs genes referred as mixed transcriptome (Wwp2, Bgn).

16) The authors propose a role of *Klf2/4* in attachment differentiation. What is the temporality in expression of *Klf2/4* vs the putative downstream factors such as Gli1, Col5a1?

17) In ISH figures, some cells in the cartilage compartment also seem to coexpress tenocyte/cartilage markers. Can the authors comment on that?

18) How did the authors adapt MARS-Seq (a single cell RNA seq pipeline taking advantage of cell sorting) to a bulk analysis? More specifically, it isn't clear how laser capture technique was combined with the MARS-seq protocol.

19) The resolution on the single molecule FISH does not allow to really appreciate a large coexpression of the presented markers in the area.

20) Figure 4: Could the authors indicate more clearly the demarcation between cartilage and connective tissue where double labeling is found?

21) It would be interesting to know if the loss of the gene expression in Figure 5 results in a morphological abnormal attachment at later postnatal stages. If the authors have looked, it would be helpful to comment in the Discussion or include the data. How much of the intermediate gene expression program in the attachment site is dependent upon Klf regulation?

22) Is this transcriptional state-sharing permanent or transitional? Their work could be nicely contrasted and compared with some studies examining transcriptional heterogeneity/the co-expression of multiple cell fates as a mechanism cells used to transition from (multipotent) progenitor states to committed fates. Enthesis tissue would be an interesting and unique situation where possibly this intermediate shared transcriptional state is maintained to generate a new cell type. Possible references for transcriptional heterogeneity in progenitors include: Soldatov et al., 2019 and Johnson et al., 2015.

23) The section on the AEG/esophagus-stomach boundary should be better integrated with their own data or removed from the Discussion. It was not clearly stated how these two tissues are similar other than being border tissues. It is recommended to expand this section to include more specific examples how these regions (enthesis and esophagus) are related. Perhaps this esophageal boundary has also been shown to have a shared transcriptional/epigenetic state with neighboring tissues?

---

## [Author Response]

Essential revisions1) The test of enhancers by transgenesis is somewhat limited in scope (only 8 tested) and yields some surprising results that can be explored further. An important request would be to make sure of the reproducibility of experiments. It is said that 5 out of 8 selected elements drove expression in the forelimb, but not how many attempts were done for the negative ones. Moreover, as reported in the transparent report form, N=1 for Col1a1 element, N=3 for Klf2 element, N=2 for Sox9 element, N=3 for Mgp element, N=4 for Col11a1. The reviewers understand that a site-directed approach is likely to be more reproducible that random insertion, it is recommended to examine at least 3 instances per element, to be certain of some surprising results. For example, although Sox9 is not expressed in hypertrophic chondrocytes, the Sox9 element drives expression in hypertrophic chondrocytes. Moreover, Sox9 and Klf2 elements drive expression in hypertrophic chondrocytes but not in AU cells. If these results are confirmed, they may cast doubt on the conclusion that co-option of enhancers is the (only) mechanism that regulates expression in AU cells.

We appreciate the reviewer’s comments about the results of the site-directed LacZ enhancer-reporter assay. The paper that describes the new site-directed transgenic analysis method that we used (termed “enSERT”, for “enhancer INSERTION”) has been published very recently (Kvon et al., 2020). This publication informs about the superior specificity and reproducibility of this new approach, as verified by extensive analyses in mouse embryos. This method, which uses CRISPR/Cas9 for site-directed integration (knock-in) of LacZ reporter transgenes at the *H11* safe-harbor locus, allows in vivo validation of enhancer activity with higher accuracy and specificity than previous, mostly random integration-based methods. Specifically, enSERT allows to achieve “an average transgenic rate of 50% for transgenes as large as 11 kb (based on >1,200 transgenic mice resulting from the injection of >150 independent transgenic constructs), compared to a 12% transgenic rate observed in conventional random transgenesis (based on >3,000 independent transgenic constructs)” (Kvon et al., 2020).

In Kvon et al. (Figure S2), the reliable reproducibility of the method was confirmed at E11.5 in a wide range of embryonic tissues, e.g. limbs, brain and heart. Typically, site-specific multi-copy transgenic insertion events resulted in the most sensitive (and reproducible) patterns of enhancer activity (see Materials and methods subsection “Tandem integration at H11” in Kvon et al.).

In our study, we leveraged the reliable nature of enSERT to validate in vivo multiple enhancer regions at E14.5 that had been predicted by our ATAC-seq experiments. The majority of tested regions indeed revealed in vivo enhancer activities with superior reproducibility in multiple tissues. For example, while mm1988 (*Klf2* element) displayed reproducible patterns in craniofacial tissues and limbs, mm1990 (Mgp element) had reproducible patterns specifically in limbs (n=3/3 for both).

Regarding the number of examined embryos, we updated and increased the number of repetitions. We examined at least 3 embryos per enhancer and, in some cases, between 6 and 11 repetitions were made. As the reviewers suggested, we have also added the transgenic reproducibility numbers (X/Y) to Figure 4. As is the standard in the field, we compare the number of embryos with sub-regionally reproducible LacZ staining (X) to the total number of transgenic embryos obtained per tested genomic element.

As shown in Author response table 1, while some of the elements, such as the *Klf2* element mm1988, displayed a high and extremely convincing reproducibly rate (N=3/3), limb-specific reproducible staining for the Col1 element mm1995 was observed in only 2 out of 9 embryos. Nevertheless, because we confirmed the limb-subregional reproducibility (on sections) in two independent biological replicates and because this element was selected based on our limb-specific ATAC-seq and ENCODE datasets of histone modification marks associated with enhancers and promoters (H3K27Ac and H3K4me1 ChIP-seq), we have reasons to believe that this result is real and represents the activity of a Col1 enhancer in attachment and tendon cells.

**Author response table 1. resptable1:** 

**Vista ID**	**N**	**Reproducible staining**	**Limb**	**Notes:**
**mm1986**	3/6	Craniofacial	Negative	
**mm1995**	2/9	Limb	Positive*	* Scored as “random integration” (Kvon et al., Cell 2020).
**mm1988**	3/3	Various	Positive	
**mm1989**	4/7	Limb	Positive	
**mm1990**	3/3	Limb	Positive	
**mm1991**	11/11	Limb, other structures with less penetrance	Positive	
**mm1992**	0/6	None	Negative	2 embryos displayed non-reproducible or background LacZ staining
**mm1994**	0/9	None	Negative	5 embryos displayed non-reproducible or background LacZ staining

One explanation for the reduced reproducibility of Col1 element may be the temporal in vivo changes in enhancer usage during development. Previous studies have shown that enhancers exhibited tightly temporally restricted predicted activity windows (Nord et al., Cell 2013). It is therefore possible that mm1995 element activity at E14.5 exemplifies the start or end of this element activity window. We therefore agree that in order to establish mm1995 as an enhancer, additional repetitions are required. Unfortunately, due to the COVID-19 situation, we do not know when this will be possible. Therefore, if the reviewers approve, we would like to keep this result in our manuscript. Obviously, we clearly state its limitation.

We fully agree with the reviewers that de novo enhancers may play a role in the regulation of attachment cells in addition to co-option of enhancers (i.e. sharing of elements with tendon or cartilage cells). This issue is discussed in detail in our next answer.

2) The authors have not systemically looked for AU enhancers that are not shared with tenocytes or chondrocytes. Combined ATAC-seq dataset and published ChIP-seq of histone marks can potentially identify new enhancers. The authors could speculate or assess if those enhancers were acquired de novo and exclusive of AU cells.

We thank the reviewer for this comment. It is important to emphasize that we do not exclude the possibility that attachment cells express a unique set of genes. However, our main purpose in this manuscript is to describe the dual fate of attachment cells. Yet, to accommodate the reviewer’s concern, we have included a list of genes that we identified to be expressed specifically in the AU. In addition, we also identified a group of intergenic elements that were accessible specifically in attachment cells. These elements may act as enhancers that drive gene expression in the attachment cells, thereby providing another level of specificity to the regulation of the development of this unique tissue. We have added a table listing these intergenic regions (newly added Supplementary file 1). Among the genes that are associated with AU-specific elements, we identified several transcription factors such as *Klf4*, *Runx3* and *Nfia*, in addition to ECM-associated genes such as *Col11a1* and *Col1a1*. As the reviewers suggested, we have added text to the Results and Discussion sections describing these findings.

3) Figure 1A, the authors present a PCA biplot : Can they be more specific on how the data were transformed prior the dimension reduction (FPKM, VST, Log transformed, CPM…?). How many genes were taken into account in the PCA; All or as it is more commonly done on the 500 most variant ones ?

The PCA shows log transformed normalized data of the 500 most variant genes. The analysis was done using a modified plotPCA function of DESeq2 (rlogTransformation with parameter blind=TRUE).

4) Figure 1 C – GO terms found are very generic, this information does not really seem to be useful. Can the authors can be more specific on the parameters they used in their GSEA analysis : test used and p-value correction (FDR q-value suggests a Benjamin and Hochberg correction, it that right ?

As the reviewer suggested, we excluded the GO analysis from the revised version of the paper. We used GOrilla to find GO annotations by loading the list of 374 genes on the background of all genes and received the GO processes that were shown in Figure 1C (please find the report with all results attached as GOPROCESS_374.txt).

5) In general, one has to give the detail on the software version (including package version) and OS type used for the bioinformatic analysis, these informations are missing from the manuscript.

As requested by the reviewer, the following information has been added to the Materials and methods section:

“The data were analyzed using the Pipeline Pilot-designed pipeline for transSeq (by INCPM, https://incpmpm.atlassian.net/wiki/spaces/PUB/pages/36405284/tranSeq+on+Pipeline-Pilot). Briefly, the analysis included adapter trimming, mapping to the mm9 genome, collapsing of reads with the same unique molecular identifiers (UMI) of 4 bases (R2) and counting of the number of reads per gene with HTseq-count (Anders et al., 2015), using the most 3’ 1000 bp of each RefSeq’s transcript. DESeq2 (version 1.4.5, Love, Huber and Anders, 2014) was used for normalization and differential expression analysis with betaPrior set to true, cooksCutoff=FALSE, independentFiltering=FALSE. Benjamini-Hochberg method was used to adjust the raw p-values for multiple testing. Genes with adjusted p-value ≤ 0.05 and fold change ≥ 2 between every two conditions were considered as differential. PCA analysis was done using log transformed normalized data (DESeq2 function rlogTransformation with parameter blind=TRUE) was done with a modified plotPCA function of DESeq2.”

6) In the sentence: "This suggests that the attachment cell transcriptome is largely shared with both chondrocytes and tenocytes (Figure 1A, PC1 52.47%)", the word largely is misleading.

The word “largely” was removed from the PCA description.

7) The authors do not discuss the variation both on the first and the second PC and of the attachment samples. This is a big issue because there are only 2 samples for this category of cells in which the intra-group variability is very high. This leads to a poor statistical parameter estimation giving rise to poor statistical test outcome.

We thank the reviewer for this comment. Indeed, we fully agree that more samples would have increased our statistical power and possibly identify more AU-specific genes. However, obtaining this tissue was very challenging technically. Nevertheless, applying adjusted p-values to these samples provided us with a list of statistically significant genes that are differentially expressed in attachment cells. Importantly, the PCA results were further validated by an extensive set of experiments, such as in situ hybridization and scRNA-Seq.

8) Legend of Figure 1: The term MARS-Seq is slightly misleading as it is usually associated with single cell RNAseq analysis. For clarity, please write instead bulk-MARS-Seq.

We have changed the term MARS-Seq to bulk-MARS-Seq in the manuscript.

9) "To further support our initial observation that the transcriptome of the attachment cells is a mixture of chondrocyte and tenocyte transcriptomes, we clustered the statistically significant differentially expressed genes between all samples into 5 clusters, using CLICK". Same remark as the use of contrasts in DESeq2.

As requested by the reviewer, this text have been added to the Materials and methods section: “Clustering of the log normalized read count of differentially expressed genes was done using click algorithm (Expander package version 7.1, Ulitsky et al., 2010), followed by visualization by R (R Core Team, 2013).”

10) In the sentence : "From these two clusters, 374 genes, 320 of them tenogenic and 54 chondrogenic, were also found to be expressed by attachment cells." It is unclear what "so found to be expressed by attachment cells" mean? For instance, for the tenogenic markers, does this mean that in the attachment vs. chondrocytes comparison these genes are up-regulated in the attachment cells? In that case, how are the tenogenic markers defined, using the tenocytes vs chondrocytes comparison? Would it be possible to have a Venn diagram to help follow the process to define the different marker identifications? Has a simpler method using contrasts in DESeq2 been tested? If yes, do the results converge with the ones presented here? Are these "statistically significant differentially expressed genes between all samples" coming from a pairwise wald-test or a likelihood ratio test? the Materials and methods suggest that the wald-test was used. Please clarify.

Genes were selected as follows: First, DESeq2 yielded 865 differential genes (i.e., between all samples, Figure 1—figure supplement 2), based on criteria that are described in Materials and methods (“Genes with adjusted p-value ≤ 0.05 and fold change ≥ 2 between every two conditions were considered as differential”). Second, to demonstrate how attachment cells function as an intermediate tissue, from the 865 differential genes we selected the genes in cluster 1 (up-regulated in tenocytes vs. chondrocytes samples, hence “tenogenic”) and cluster 2 (up-regulated in chondrocytes vs. tenocytes samples, hence “chondrogenic”). Third, we averaged the normalized number of reads in attachment samples of each gene on this list and selected genes with 30 normalized reads or more. Thus, the 374 genes that are shown in Figure 1 represent the activation of both tenogenic and chondrogenic gene expression in E14.5 attachment cells. We have added this description to the Materials and methods section.

11) In a previous study the authors investigated the emergence of the attachment unit (AU) with focus on bone eminence progenitors (co-expressing Sox9 and SCX up to E12.5 and expressed Col2 after E12.5 according to Col2a1CreERT2 lineage tracing). Here, they focus on the transcriptome of E14.5 attachment cells from the deltoid tuberosity, however these appear to be different from tuberosity progenitors (adjacent chondrocytes) as described in Figure 1—figure supplement 1. A better definition of what is defined as attachment unit in this paper vs previous papers and/or the AU subcompartments would help clarify the populations that are being examined.

Indeed, the reviewer accurately cites our previous paper and we understand how the use of AU in both papers can lead to some confusion regarding its definition. We have added to the Introduction a paragraph that explains the development of the AU, serving as the enthesis primordium, whose cells co-express *Sox9* and Scx. As development proceeds, this field is divided into sub compartments, namely the cartilaginous bone eminence (Col2-positive) on one side, the tendon (Scx-positive) on the other side and, connecting between them, the attachment cells, which will eventually give rise to the enthesis. The objective of this paper is to address the understudied differentiation sequence of this third population.

12) Moreover, as the constitutive Col2a1Cre did not label the AU, but in the previous study did label the AU/bone eminence progenitors, it is unclear what the exact definition of AU is.

In the previous study, as in this work, the constitutive Col2a1Cre labeled the bone side of the AU, namely the bone eminence. In this work, the term AU refers to the tissue that forms between tendon and cartilage, as explained in the previous response.

13) When using the Col2a1-Cre, R26R-tdTomato and Scx-GFP, the authors mention : Unexpectedly, the two reporters failed to label the attachment cells that were located in between these two populations. This failure might be due to a missing regulatory element in one of the constructs that was used to produce each transgenic reporter. However, in Figure 1—figure supplement 1, subpopulation 5 seems to have SCX+ cells. Is this an error of labelling? What is the orientation of this section?

The orientation of this section is transverse. When observing a series of transverse sections from E14.5 forelimbs, we identified in each section Col2+ labeled cells, Scx-GFP+ labeled cells and, between them, cells that were unlabeled. Figure 1—figure supplement 1 is one of many collected sections in which subpopulation 5 is of unlabeled cells. Regarding GFP-labelled cells at the margins of the laser-captured area, to maximize the size of extracted tissue, we stretched the borders of the AU all the way to the flanking Scx-GFP+ cells. When the laser cuts the piece of tissue it burns the boundaries, so the RNA in cells located at the boundaries is degraded.

14) For FACS and ATACseq analysis the authors use Sox9CreERT2;tdTomato;SCX-GFP and Col2CreERT2;tdTomato;SCX-GFP. It is not clear why for FACS the Col2CREERT2 line is used while for LCM the constitutive one is used. Moreover, as previously reported, they isolate attachment cells as double positive SOX9/SCX cells. Here again, do the cells taken for analysis include those of the tuberosity itself? Col2CreERT2 with Tamox at E12.5 should be labeling the tuberosity too. It is unclear which cells from which Cre/reporter combination have been used for the ATACseq experiments of Figure 3.

When we initiated the LCM experiment, we assumed that Col2 would label the attachment cells. However, to our surprise, attachment cells were Col2-negative. Nevertheless, as the Scx and Col2 reporters clearly defined the boundaries between cell types, we could still use this line for LCM. This part is explained in detail in the Materials and methods section.

Because for the ATAC-seq experiment we needed to isolate the cells using FACS, we had to change our labeling strategy. Therefore, we used a combination of *Sox9*-CreER and Scx-GFP mice. This indeed worked well for tendon and attachment cells, but the number of isolated chondrocytes was insufficient. Therefore, we used the Col2-CreER^T^ line to label chondrocytes and isolate them. To clarify this issue, we have added text to the Results section.

As the reviewer noted, this labeling should also label the chondrocytes that compose the tuberosity. However, due to the large difference in cell numbers between the tuberosity and the rest of the cartilage that was collected, we think that this had a limited (if any) effect on our results.

15) For Figure 2, a scheme showing where exactly in the bone we are located and how it has been sectioned would be helpful. Also, it would be nice to perform single molecule FISH on top of Col2Cre:R26TOM:SCX lineage tracing to show the specificity of the colocalization in the "double reporter-negative" area. Also, including the KLF2/4 FISH at this point would help visualize distinctions between genes belonging to cluster 5 (unique to AU) vs genes referred as mixed transcriptome (Wwp2, Bgn).

We have added to Figure 1 a scheme showing how we sectioned the deltoid and great tuberosities for FISH. Using E14.5 ScxGFP-Col2Cre-tdTomato transverse sections for Bgn and Wwp2 double smFISH (conjugated to Quasar 670 and CAL Fluor Red 610, respectively) is problematic, since tdTomato (540nm-581nm) is a very strong reporter that may mask the signal of the Wwp2 probe coupled to Quazar 610 (590nm-610nm), due to a "bleed through" of the tdTomato into the Quazar610 filter.

As an alternative, to address the reviewer’s comment we performed a 10x genomics experiment on *Sox9*+Scx+ cells, where we could search for the transcripts of Bgn and Wwp2 in addition to expression of *Klf2*/*Klf4* in *Sox9*+Scx+ cells. Figure 2A,B summarizes this analysis. In short, this experiment exemplifies that *Sox9*+Scx+ cells indeed express both chondrogenic and tenogenic markers, in addition to KLF’s (Figure 5B).

16) The authors propose a role of Klf2/4 in attachment differentiation. What is the temporality in expression of Klf2/4 vs the putative downstream factors such as Gli1, Col5a1?

To address the temporality in gene expression, we performed transcriptome analysis (bulk MARS-Seq) of FACS-sorted attachment cells E13.5, when it is first possible anatomically to identify the attachment site (*Sox9*+Scx+ cells, n=3 for each gene; see Author response table 2). Results showed that at that time point, both *Klf2* and *Klf4*, as well as Col5a1 and Gli1, are already expressed. Since our analysis of the *Klf2/4* double mutants clearly showed a reduction (Figure 5) in expression of Gli1 and Col5a1, *bona fide* markers of the differentiating enthesis (Felsenthal et al., 2018; Schwartz et al., 2015), we concluded that *Klf2* and *Klf4* are necessary for attachment cell differentiation.

**Author response table 2. resptable2:** 

Gene	Bulk MARS-Seq normalized reads (mean)
Klf2	203.67
Klf4	384.75
Gli1	349.15
Col5a1	391.62

17) In ISH figures, some cells in the cartilage compartment also seem to coexpress tenocyte/cartilage markers. Can the authors comment on that?

We agree with the reviewer that in some cases, at the borders the expression patterns of tendon and cartilage markers are not sharply segregated. This is mainly because during development, the borders between the different tissues that compose the attachment site are not sharp and clear, as the process of compartmentalization is still underway. Nevertheless, we believe that in general, expression of ISH markers followed the expected spatial patterns. The scRNA-seq results further support this conclusion.

18) How did the authors adapt MARS-Seq (a single cell RNA seq pipeline taking advantage of cell sorting) to a bulk analysis? More specifically, it isn't clear how laser capture technique was combined with the MARS-seq protocol.

MARS-Seq was indeed originally developed as a FACS-based method for single-cell RNA sequencing [69] (Jaitin et al., 2014; Keren-Shaul, 2019). Due to the low RNA input in scRNA-seq, the authors further developed a bulk MARS-seq protocol, which is an adaptation of the single-cell MARS-seq. The input for bulk MARS-seq is clean RNA and hence it can be used for expression profiling by purifying RNA from LCM-isolated samples, FACS-sorted cells and more.

As explained in detail in the Materials and methods subsection “Laser capture microdissection”, following LCM of E14.5 cryo-sections of the attachment site, we purified the RNA using an RNA purification kit (RNeasy FFPE Kit, Qiagen). The resulting RNA was the input for the bulk MARS-seq protocol. Briefly, RNA from each sample was barcoded during reverse transcription and pooled. Following Agencourct Ampure XP beads cleanup (Beckman Coulter), the pooled samples underwent second strand synthesis and were linearly amplified by T7 in vitro transcription. The resulting RNA was fragmented and converted into a sequencing-ready library by tagging the samples with Illumina sequences during ligation, RT, and PCR. Libraries were quantified by Qubit and TapeStation as well as by qPCR for actb gene as previously described (Jaitin et al., 2014; Keren-Shaul et al., 2019). Sequencing was done on a Hiseq 2500 SR50 cycles kit (Illumina).

19) The resolution on the single molecule FISH does not allow to really appreciate a large coexpression of the presented markers in the area.

Figure 2C, showing the smFISH of biglycan and *Wwp2* in E14.5 attachment cells, was increased in resolution to easily appreciate the coexpression of the markers.

20) Figure 4: Could the authors indicate more clearly the demarcation between cartilage and connective tissue where double labeling is found?

We have added clearer demarcation between cartilage and connective tissue.

21) It would be interesting to know if the loss of the gene expression in Figure 5 results in a morphological abnormal attachment at later postnatal stages. If the authors have looked, it would be helpful to comment in the Discussion or include the data. How much of the intermediate gene expression program in the attachment site is dependent upon Klf regulation?

We agree with the reviewers. However, due to postnatal lethality, our analysis of E18.5 is the latest time point that can be studied in this mouse line.

22) Is this transcriptional state-sharing permanent or transitional? Their work could be nicely contrasted and compared with some studies examining transcriptional heterogeneity/the co-expression of multiple cell fates as a mechanism cells used to transition from (multipotent) progenitor states to committed fates. Enthesis tissue would be an interesting and unique situation where possibly this intermediate shared transcriptional state is maintained to generate a new cell type. Possible references for transcriptional heterogeneity in progenitors include: Soldatov et al., 2019 and Johnson et al., 2015.

As suggested, we have added a paragraph to the Discussion regarding this issue.

23) The section on the AEG/esophagus-stomach boundary should be better integrated with their own data or removed from the Discussion. It was not clearly stated how these two tissues are similar other than being border tissues. It is recommended to expand this section to include more specific examples how these regions (enthesis and esophagus) are related. Perhaps this esophageal boundary has also been shown to have a shared transcriptional/epigenetic state with neighboring tissues?

This part of the Discussion was edited to fit with the reviewer requirements.